# An open-source, high-performance tool for automated sleep staging

Raphael Vallat*, Matthew P Walker*

Center for Human Sleep Science, Department of Psychology, University of California, Berkeley, Berkeley, United States

**Abstract** The clinical and societal measurement of human sleep has increased exponentially in recent years. However, unlike other fields of medical analysis that have become highly automated, basic and clinical sleep research still relies on human visual scoring. Such human-based evaluations are time-consuming, tedious, and can be prone to subjective bias. Here, we describe a novel algorithm trained and validated on +30,000 hr of polysomnographic sleep recordings across heterogeneous populations around the world. This tool offers high sleep-staging accuracy that matches human scoring accuracy and interscorer agreement no matter the population kind. The software is designed to be especially easy to use, computationally low-demanding, open source, and free. Our hope is that this software facilitates the broad adoption of an industry-standard automated sleep staging software package.

## Introduction

Sleep is fundamental to human health. Adequate sleep supports a panoply of physiological body functions, including immune, metabolic, and cardiovascular systems (*Besedovsky et al., 2019*; *Cappuccio and Miller, 2017*; *Harding et al., 2020*). Within the brain, sufficient sleep facilitates optimal learning, memory, attention, mood, and decision-making processes (*Ben Simon et al., 2020*; *Walker, 2009*). Improving sleep health has therefore emerged as a preventive strategy to reduce the risk of cardiovascular and metabolic disease, all-cause mortality risk, and more recently, the accumulation of Alzheimer's disease pathology within the brain (*Cappuccio et al., 2010*; *Leary et al., 2020Cappuccio and Miller, 2017*; *Ju et al., 2014*; *Winer et al., 2020*). Considering this impact, the demand for quantifying human sleep at a research-, clinical-, and consumer-based level has increased expeditiously over the past decade (*Fleming et al., 2015*; *Shelgikar et al., 2016*).

Polysomnography (PSG) – the simultaneous measurement of brainwaves, eye movements, muscle activity, heart rate, and respiration – is the gold standard for objective physiological quantification of human sleep. The classification of sleep stages across the night provides information on the overall architecture of sleep across the night, as well as the duration and proportion of the sleep stages, all of which inform the diagnosis of sleep disorders and specific diseases states.

Currently, such sleep scoring is typically performed by humans, accomplished by first dividing the night of PSG recording into 30 s segments (called 'epochs'). Each epoch is then assigned a sleep stage based on standard rules defined by the American Academy of Sleep Medicine (AASM, *Berry et al., 2012*; *Iber et al., 2007*).

Every night, thousands of hours of sleep are recorded in research, clinical, and commercial ventures across the globe. However, since sleep staging is performed visually by human experts, it represents a pragmatic bottleneck. It is a non-trivial, time-intensive process, with visual scoring of a single night of human sleep requiring up to 2 hr to complete by a well-trained individual. As if not more critical, this human-scored approach suffers from issues of lower than desirable *inter*scorer consistency of agreement (~83% agreement, *Rosenberg and Van Hout, 2013*). Moreover, the same scoring individual typically experiences low *intra*scorer agreement of the same sleep recording (~90%, *Fiorillo et al.,*

*For correspondence:
raphaelvallat@berkeley.edu (RV);
mpwalker@berkeley.edu (MPW)

Competing interest: The authors declare that no competing interests exist.

*2019*). That is, different human sleep-scoring experts presented with the same recording will likely end up with somewhat dissimilar sleep-staging evaluations, and even the same expert presented with the same recording assessed at two different time points will arrive at differing results.

Advances in machine learning have motivated efforts to try and classify sleep using automated systems. Several such automatic sleep-staging algorithms have emerged in recent years. While an exhaustive review of such sleep-staging algorithms is beyond the scope of this article, we offer an overview of some of the most relevant algorithms within the last 5 years. For a more in-depth review, we refer the reader to *Fiorillo et al., 2019*.

*Sun et al., 2017* reported an algorithm that was trained and evaluated on 2000 PSG recordings from a single-sleep clinic. The overall Cohen's kappa on the testing set was 0.68 (n = 1000 PSG nights). Thereafter, and algorithm termed 'Z3Score' (*Patanaik et al., 2018*) was published, trained and evaluated on ~1700 PSG recordings from four datasets. Here, the overall accuracy ranged from 89.8% in healthy adults/adolescents to 72.1% in patients with Parkinson's disease. The freely available 'Stanford-stage' algorithm (*Stephansen et al., 2018*) was trained and evaluated on 10 clinical cohorts (~3000 recordings), and presented with an accuracy of 87% against consensus human sleep stage scoring. A year later, the 'SeqSleepNet' algorithm (*Phan et al., 2019*) was published, which represented a method trained and tested using a 20-fold cross-validation of 200 nights of PSG data, with an overall accuracy of 87.1%. Finally, the recent U-Sleep algorithm (*Perslev et al., 2021*) was established using PSG recordings from 15,660 participants of 16 clinical studies. The overall accuracy was not reported, though the mean F1-score—a measure reflecting the quality of the prediction against the consensus scoring of five human experts—was 0.79 for healthy adults and 0.76 for patients with sleep apnea.

Despite this growing number of automated tools, accurate automated sleep staging has yet to become a de facto standard in the field. There are likely several reasons for this. First, some algorithms are not free, sitting behind paywalls and/or not publicly available. Second, other algorithms require paid software to run the nonetheless free algorithm on, such as MATLAB. Third, some algorithms have been trained on a sample size too small for robustness and/or data from a single-sleep center or population. As a result, their ability to generalize to other recording systems and/or populations, including patients with sleep disorders or across broad age ranges, has been a concern. Fourth, setting up and running these algorithms is often too complicated for most typical individuals as they require moderate- to high-level programing experience, creating a barrier of entry to adoption and broad use. A final common limitation to all these algorithms is that they were evaluated on different testing datasets and using different metrics, which prevents a direct comparison of the performance of these algorithms and had lead to understandable confusion among some sleep researchers and clinicians.

Seeking to address all such issues, here, we describe a free, flexible, and easy-to-use automated sleep-staging software that has been trained and evaluated on more than 30,000 hr of PSG-staged sleep from numerous independent and heterogeneous datasets with a wide range of age, ethnicities, and health status levels. Furthermore, we test our algorithm against two recent sleep-staging algorithms (*Perslev et al., 2021*; *Stephansen et al., 2018*) on a previously unseen consensus-scored sleep dataset of healthy adults and patients with obstructive sleep apnea (OSA).

## Results
### Descriptive statistics
The *training set* consisted of more than 31,000 hr of PSG data across 3163 unique full-night PSG recordings from seven different datasets (Cleveland Children's Sleep and Health Study [CCSHS], n = 414; Cleveland Family Study [CFS], n = 586; Childhood Adenotonsillectomy Trial [CHAT], n = 351; Home Positive Airway Pressure [HomePAP], n = 82; Multi-Ethnic Study of Atherosclerosis [MESA], n = 575; Osteoporotic Fractures in Men Study [MrOS], n = 565; Sleep Heart Health Study [SHHS], n = 590). All these datasets are publicly available from the National Sleep Research Resource (NSRR) website (http://sleepdata.org). In total, these 3163 training nights represent almost 4 million unique 30 s epochs. Demographics and health data of the training set are shown in *Table 1*. The average apnea-hypopnea index (AHI) was 12.9 ± 16.35 (median = 6.95, range = 0–125). The AHI was ≥15 (= moderate sleep apnea) for 29% of the nights. The MESA dataset had the highest average AHI (19.2 ± 18.1), while the CCSHS had the lowest AHI (1.5 ± 5.2).

**Table 1.** Demographics of the training set and testing set 1.

Age, body mass index (BMI), and apnea-hypopnea index (AHI) are expressed in mean ± standard deviation. The p-value for these three variables was calculated using a Welch's two-sided t-test, and the effect size refers to the Hedges g. All the other categorical variables are expressed in percentage. Significance was assessed using a chi-square test of independence, and effect size refers to the Cramer's V. The apnea severity was classified as follows: none = AHI < 5 per hour, mild = AHI ≥ 5 but < 15, moderate = AHI ≥ 15 but < 30, and severe = AHI ≥ 30. p-Values were not adjusted for multiple comparisons.

| | Training | Testing set 1 | p-Value | Effect size |
|---|---|---|---|---|
| No. nights | 561 | 3163 | - | - |
| Age | 49.79 ± 26.38 | 45.25 ± 28.18 | **<0.001** | 0.170 |
| BMI | 27.65 ± 7.40 | 26.83 ± 7.30 | **0.013** | 0.111 |
| Sex (% male) | 56.56 | 55.385 | 0.630 | 0.008 |
| *Race (%)* | - | - | 0.985 | 0.006 |
| Black | 29.47 | 29.23 | - | - |
| White | 57.92 | 58.46 | - | - |
| Hispanic | 7.24 | 6.84 | - | - |
| Asian or Pacific Islander | 5.375 | 5.47 | - | - |
| AHI (events/hour) | 12.94 ± 16.35 | 11.99 ± 14.99 | 0.166 | 0.059 |
| *Apnea severity* | - | - | 0.820 | 0.016 |
| None | 42.78 | 43.76 | - | - |
| Mild | 27.79 | 27.18 | - | - |
| Moderate | 17.325 | 18.12 | - | - |
| Severe | 12.11 | 10.94 | - | - |
| Insomnia (%) | 6.95 | 4.50 | 0.255 | 0.019 |
| Depression (%) | 15.83 | 13.158 | 0.407 | 0.014 |
| Diabetes (%) | 15.82 | 17.99 | 0.260 | 0.018 |
| Hypertension (%) | 38.83 | 35.27 | 0.152 | 0.023 |

The *testing set 1* consisted of 585 unique full-night PSG recordings from six different datasets (CCSHS, n = 100; CFS, n = 99; CHAT, n = 100; MESA, n = 97; MrOS, n = 90; SHHS, n = 99). The testing nights were randomly selected from a subset of nights that were not included in the training set (see Materials and methods). We did not include any nights from the HomePAP dataset because the latter only consisted of 82 nights after preprocessing, all of which were used for training. In total, these 585 testing nights represent 716,367 unique 30 s epochs, or roughly 6000 hr of PSG data. Demographics and health data of the testing set one are shown in *Table 1*. Importantly, there was no significant difference in the sex ratio, race distribution, or in the proportion of individuals diagnosed with insomnia, depression, or diabetes. Furthermore, there was no difference in AHI and the proportion of individuals with minimal, mild, moderate, or severe sleep apnea. Although participants were randomly assigned in the training/testing sets, there were some significant differences between these two sets. Specifically, age and body mass index (BMI) were lower in the testing set compared to the original training set (p's<0.05), although the effect sizes of these differences were all below 0.17 (considered small or negligible).

The *testing set 2* consisted of 80 full-night PSG recordings from the publicly available Dreem Open Dataset (DOD) dataset (*Guillot et al., 2020*). The DOD consists of 25 nights from healthy adults (DOD-Healthy: age = 35.3 ± 7.51 years, 76% male, BMI = 23.8 ± 3.4) and 55 nights from patients with

OSA (DOD-Obstructive: age = 45.6 ± 16.5 years, 64% male, BMI = 29.6 ± 6.4, AHI = 18.5 ± 16.2). Individual-level demographics and medical history were not provided for the DOD datasets; group averages for age, BMI, and AHI are reported from *Guillot et al., 2020*. Five nights were excluded because the duration of the PSG data and the duration of the hypnogram differed by more than 1 min, leading to a final sample of 75 nights (~607 hr of data). Each night of the testing set 2 was scored by five registered sleep technicians, thus allowing to test the algorithm against a consensus of human scorers (see Materials and methods). Unlike the NSRR datasets, which were split into training and testing subsets, the two DOD datasets (healthy and patients) were only used for performance evaluation. This provides a more robust estimation of the algorithm's performance on real-world data from a new sleep clinic or laboratory.

## Validation results

### Testing set 1: NSRR holdout

Overall performance of the algorithm on the testing set 1 is described in *Figure 1A*. Median accuracy, calculated across all 585 testing nights, was 87.46% . The median Cohen's kappa was 0.819, indicating excellent agreement, and similarly, the median Matthews correlation coefficient was 0.821. The CCSHS database testing set had the highest overall accuracy (median = 90.44%), while the MESA database had the lowest accuracy (median = 83.99%). In addition to traditional sleep staging, the algorithm is able to quantify the probability of each sleep stage at each 30 s epoch, which can then be used to derive a confidence score at each epoch. The median confidence of the algorithm across all the testing nights was 85.79% . Nights with a higher average confidence had significantly higher accuracy (*Figure 1B*, $r = 0.76$, $p<0.001$).

Next, we tested the classification performance of individual sleep stages (*Figure 1C*). The overall sensitivity (i.e., the percent of epochs that were correctly classified, see Materials and methods) of N3 sleep was 83.2%, leading to a median F1-score across all testing nights of 0.835. Rapid eye movement (REM) sleep, N2 sleep, and wakefulness all showed sensitivities above 85% (all median F1-scores ≥ 0.86). N1 sleep showed the lowest agreement, with an overall sensitivity of 45.4% and a median F1-score of 0.432 – a finding consistent with this sleep stage often showing very low agreement across human sleep scorers (<45%, *Malhotra et al., 2013*). These algorithm performance results were consistent when examined separately for each database (*Figure 1—figure supplement 1*).

Further examination of the confusion matrix showed that the two most common inaccuracies of the algorithm were (1) mislabeling N1 sleep as N2 sleep (27.5% of all true N1 sleep epochs) and (2) mislabeling N3 sleep as N2 sleep (16.1% of all true N3 sleep epochs). Importantly, the algorithm made few blatant errors, such as mislabeling N3 sleep as REM sleep (0.2%) or mislabeling REM sleep as N3 sleep (0.03%). In addition, when the algorithm misclassified an epoch, the second highest probability stage predicted by the algorithm was the correct one in 76.3% ± 12.6% of the time.

Additional analyses indicated that the algorithm was more prone to inaccuracies around sleep-stage transitions than during stable epochs (*Figure 2F*; mean accuracy around a stage transition: 69.15% ± 7.23%, during stable periods: 94.08% ± 5.61%, $p<0.001$). Similarly, and as expected, accuracy was significantly greater for epochs flagged by the algorithm as high confidence (≥80% confidence) than in epochs with a confidence below 80% (95.90% ± 3.70% and 62.91% ± 6.6%, respectively, $p<0.001$). The distribution of confidence levels across sleep stages is shown in *Figure 1—figure supplement 2*. The algorithm was most confident in epochs scored as wakefulness by the human scorer (mean confidence across all epochs = 92.7%) and least confident in epochs scored as N1 sleep (mean = 63.2%).

We further tested for any systematic bias of the algorithm towards one or more sleep stages. The overall proportion of each sleep stage in the testing nights was similar between the human and automatic staging (all Cohen's d < 0.11; *Figure 1D*), indicating a negligible bias. Similarly, the percentage of stage transitions across the night was consistent in the human and automatic scoring (mean ± STD: 11.12% ± 5.19% and 11.46% ± 4.11%, respectively, Cohen's d = 0.07).

### Testing set 2: Consensus scoring on the DOD-Healthy and DOD-Obstructive datasets

Next, we examined the performance of YASA on the testing set 2, a previously unseen dataset of healthy and sleep-disordered breathing patients that was scored by five registered experts (*Guillot*

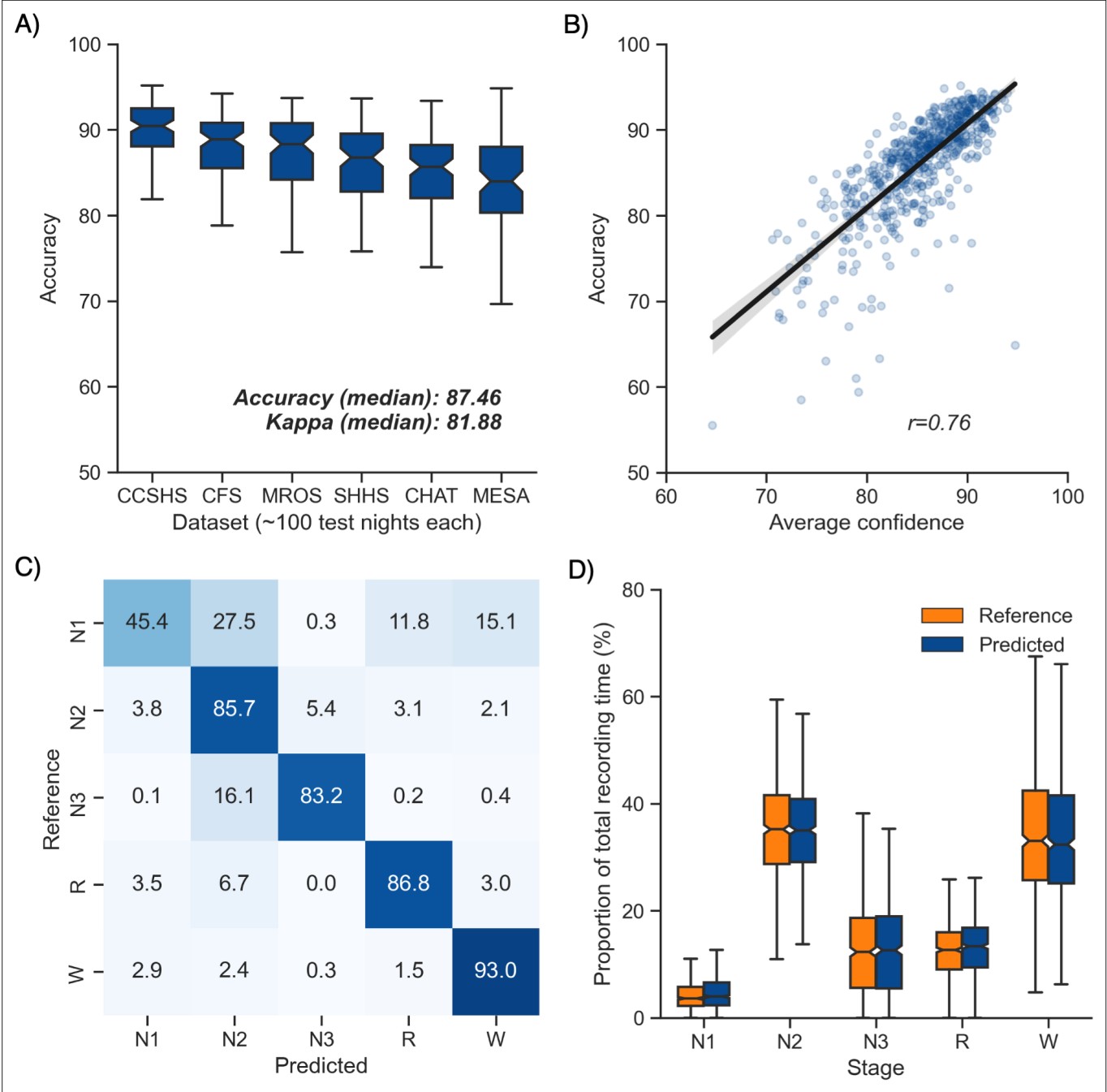

**Figure 1.** Performance of the algorithm on the testing set 1 (n = 585 nights). (**A**) Accuracy of all the testing nights, stratified by dataset. The median accuracy across all testing nights was 87.5%. (**B**) Correlation between accuracy and average confidence levels (in %) of the algorithm. The overall confidence was calculated for each night by averaging the confidence levels across all epochs. (**C**) Confusion matrix. The diagonal elements represent the percentage of epochs that were correctly classified by the algorithm (also known as sensitivity or recall), whereas the off-diagonal elements show the percentage of epochs that were mislabeled by the algorithm. (**D**) Duration of each stage in the human (red) and automatic scoring (green), calculated for each unique testing night and expressed as a proportion of the total duration of the polysomnography (PSG) recording.

The online version of this article includes the following figure supplement(s) for figure 1:

**Figure supplement 1.** Confusion matrices of the testing set 1, stratified by dataset.

**Figure supplement 2.** Distribution of confidence scores across sleep stages in the testing set 1.

**Figure supplement 3.** Performance of the YASA, *Stephansen et al., 2018*, and *Perslev et al., 2021* algorithms on the DOD-Healthy testing set (n = 25 healthy adults).

*Figure 1 continued on next page*

*Figure 1 continued*

**Figure supplement 4.** Confusion matrices of each individual human scorer on the DOD-Healthy testing set (n = 25 healthy adults).

**Figure supplement 5.** Performance of the YASA, *Stephansen et al., 2018*, and *Perslev et al., 2021* algorithms on the DOD-Obstructive testing set (n = 50 patients with obstructive sleep apnea).

**Figure supplement 6.** Confusion matrices of each individual human scorer on the DOD-Obstructive testing set (n = 50 patients with sleep apnea).

**Figure supplement 7.** Top 20 most important features of the classifier.

*et al., 2020*; *Perslev et al., 2021*). Performance of the algorithm is reported in *Table 2* for healthy adults (n = 25) and *Table 3* for individuals with sleep disorders.

In healthy adults, the median accuracy of YASA against the consensus scoring of the five experts was 86.6% . The median kappa across all nights was 80.1%, indicating excellent agreement. Median and interquartile range of F1-scores for each sleep stage across all nights are reported in *Table 2*. We then compared the performance of YASA against each of the five individual human scorers, as well as against two recently published sleep-staging algorithms (see Materials and methods, *Perslev et al., 2021*; *Stephansen et al., 2018*). Pairwise t-tests adjusted for multiple comparisons using the Holm

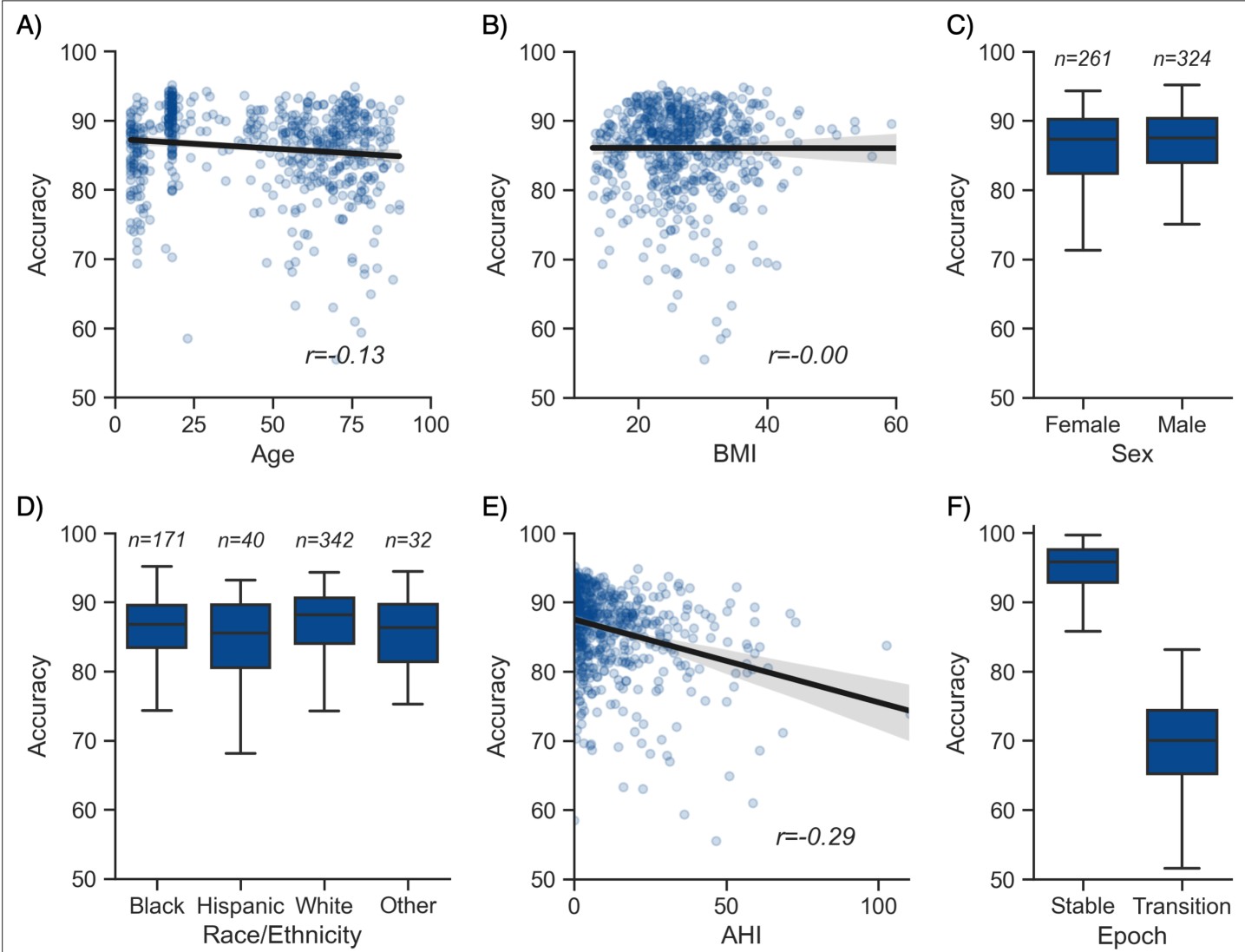

**Figure 2.** Moderator analyses. Accuracy of the testing nights as a function of age (**A**), body mass index (BMI) (**B**), sex (**C**), race (**D**), apnea-hypopnea index (AHI) (**E**), and whether or not the epoch is around a stage transition (**F**). An epoch is considered around a transition if a stage transition, as defined by the human scoring, is present within the 3 min around the epoch (1.5 min before, 1.5 min after).

**Table 2.** Comparison of YASA against two existing algorithms and individual human scorers on the DOD-Healthy dataset (healthy adults, n = 25).

Values represent median ± interquartile range across all n = 25 nights. The YASA column shows the performance of the current algorithm against the consensus scoring of the five human experts (see Materials and methods). The *Stephansen et al., 2018* and *Perslev et al., 2021* columns show the performance of two recent deep-learning-based sleep-staging algorithms (*Perslev et al., 2021*; *Stephansen et al., 2018*). The H1–H5 columns show the performance of each individual human scorer against an unbiased consensus (see Materials and methods). Asterisks indicate significant differences with YASA. p-Values were adjusted for multiple comparisons row-wise using the Holm method. Accuracy is defined as the overall agreement between the predicted and ground-truth sleep stages. F1 is the F1-score, calculated separately for each sleep stage. F1-macro is the average of the F1-scores of all sleep stages.

| | YASA | Stephansen et al., 2018 | Perslev et al., 2021 | H1 | H2 | H3 | H4 | H5 |
|---|---|---|---|---|---|---|---|---|
| Accuracy | 86.6 ± 6.2 | 86.4 ± 5.5 | 89.0 ± 4.8 | 86.5 ± 7.9 | 86.3 ± 5.9 | 86.6 ± 5.9 | 78.8 ± 9.2* | 86.3 ± 7.2 |
| F1 N1 | 52.0 ± 21.6 | 50.0 ± 19.9 | 61.8 ± 18.0* | 50.0 ± 12.2 | 53.7 ± 18.1 | 53.2 ± 19.8 | 38.7 ± 22.1* | 51.7 ± 19.2 |
| F1 N2 | 88.5 ± 5.3 | 89.5 ± 6.0 | 89.1 ± 5.7 | 89.1 ± 5.6 | 88.4 ± 6.0 | 88.1 ± 6.6 | 83.3 ± 5.2* | 88.3 ± 6.2 |
| F1 N3 | 87.0 ± 9.7 | 81.1 ± 22.5 | 84.6 ± 14.6 | 86.9 ± 17.2 | 81.4 ± 18.2 | 82.1 ± 20.6 | 82.5 ± 20.3 | 84.0 ± 11.7 |
| F1 REM | 92.6 ± 9.7 | 91.9 ± 5.6 | 94.4 ± 5.1* | 88.0 ± 13.3 | 90.6 ± 5.5 | 92.7 ± 7.0* | 90.0 ± 9.2 | 92.7 ± 8.1 |
| F1 WAKE | 83.9 ± 10.3 | 87.0 ± 12.2 | 90.5 ± 9.5* | 83.7 ± 9.5 | 87.4 ± 12.0 | 85.7 ± 9.2 | 78.1 ± 28.9 | 84.4 ± 16.8 |
| F1 macro | 78.5 ± 9.4 | 79.0 ± 8.5 | 82.7 ± 7.7* | 78.0 ± 9.0 | 80.0 ± 9.1 | 79.5 ± 7.6 | 73.0 ± 9.0* | 79.5 ± 11.9 |

method revealed that the accuracy of YASA on this healthy validation dataset was not significantly different from the two other deep-learning-based sleep-staging algorithms or any of the individual scorers (except scorer 4, see below or *Table 2*). The F1-scores for each sleep stage were not significantly different between YASA, the *Stephansen et al., 2018* algorithm, and the human scorers 1, 2, 3, and 5. Scorer 4 had a significantly lower F1-score for N1 (p=0.034) and N2 (p<0.001), resulting in a lower overall accuracy (p=0.001). The *Perslev et al., 2021* algorithm had significantly higher F1-scores than YASA for N1, REM sleep and wakefulness (p=0.006, p<0.001, and p=0.042, respectively). F1-scores for N2 and N3 were not statistically different between YASA and the *Perslev et al., 2021* algorithm. Hypnograms of the consensus scoring and the three automated algorithms for each night of the DOD-Healthy dataset can be found in *Supplementary file 1*, with nights ranked in descending order of agreement between YASA and the consensus scoring. Confusion matrices for each scorer (human and algorithm) can be found in *Figure 1—figure supplements 3 and 4*.

The same analysis was performed in 55 patients with OSA (DOD-Obstructive). The median accuracy of YASA against the consensus scoring of the five experts was 84.3%, with a median kappa of 76.5% . Median and interquartile range of F1-scores for each sleep stage across all nights are reported in *Table 3*. Pairwise comparisons of accuracy showed that YASA was not significantly different from the *Stephansen et al., 2018* algorithm or human scorers 1, 3, 4, and 5. Human scorer 2 had a significantly lower accuracy (p=0.004), while the Perslev algorithm had a significantly higher accuracy (p=0.009). Pairwise comparisons of F1-scores showed that YASA outperformed the *Stephansen et al., 2018* algorithm and four out of five scorers for N3 sleep (all p's<0.01). However, the *Perslev et al., 2021* algorithm had significantly higher scores than YASA for N1, REM and wakefulness (all p's<0.011). Hypnograms of the consensus scoring and the three automated algorithms, ranked in descending order of agreement between YASA and the consensus scoring, can be found in *Supplementary file 2*. Confusion matrices are reported in *Figure 1—figure supplements 5 and 6*.

Additional analyses on the combined dataset (n = 75 nights, healthy and patients combined) revealed that, first, the accuracy of YASA against the consensus scoring was significantly higher during stable epochs compared to transition epochs (mean ± STD: 91.2 ± 7.8 vs. 68.95 ± 7.7, p<0.001), or in epochs that were marked as high confidence by the algorithm (93.9 ± 6.0 vs. 64.1 ± 7.0 for low-confidence epochs, p<0.001). Second, epochs with unanimous consensus from the five human experts were four times more likely to occur during stable epochs than transition epochs (46.2% ± 12.1% of

**Table 3.** Comparison of YASA against two existing algorithms and individual human scorers on the DOD-Obstructive dataset (patients with obstructive sleep apnea, n = 50).

Values represent median ± interquartile range across all n = 50 nights. The YASA column shows the performance of the current algorithm against the consensus scoring of the five human experts (see Materials and methods). The *Stephansen et al., 2018* and *Perslev et al., 2021* columns show the performance of two recent deep-learning-based sleep-staging algorithms (*Perslev et al., 2021*; *Stephansen et al., 2018*). The H1–H5 columns show the performance of each individual human scorer against an unbiased consensus (see Materials and methods). Asterisks indicate significant differences with YASA. p-Values were adjusted for multiple comparisons row-wise using the Holm method. Accuracy is defined as the overall agreement between the predicted and ground-truth sleep stages. F1 is the F1-score, calculated separately for each sleep stage. F1-macro is the average of the F1-scores of all sleep stages.

| | YASA | Stephansen et al., 2018 | Perslev et al., 2021 | H1 | H2 | H3 | H4 | H5 |
|---|---|---|---|---|---|---|---|---|
| Accuracy | 84.30 ± 7.85 | 84.97 ± 9.48 | 86.69 ± 8.69* | 83.94 ± 12.72 | 82.51 ± 9.49* | 80.38 ± 11.99 | 82.18 ± 9.26 | 84.50 ± 9.78 |
| F1 N1 | 39.17 ± 18.17 | 41.48 ± 17.05 | 50.17 ± 21.17* | 34.44 ± 19.93 | 40.53 ± 29.96 | 39.01 ± 19.78 | 41.54 ± 21.04 | 46.19 ± 20.07* |
| F1 N2 | 87.10 ± 6.55 | 88.36 ± 9.39 | 87.26 ± 8.77 | 86.80 ± 15.96* | 83.07 ± 7.19* | 83.39 ± 11.34 | 84.63 ± 7.57 | 87.29 ± 8.45 |
| F1 N3 | 77.88 ± 31.12 | 57.16 ± 82.16* | 79.51 ± 31.28 | 71.51 ± 56.26* | 65.53 ± 45.41* | 49.97 ± 63.95* | 62.69 ± 61.22* | 74.74 ± 41.19 |
| F1 REM | 88.87 ± 10.79 | 92.50 ± 7.81 | 93.90 ± 5.40* | 89.86 ± 19.43 | 92.48 ± 9.91 | 91.26 ± 12.21 | 89.26 ± 9.83 | 91.77 ± 11.28 |
| F1 WAKE | 86.90 ± 8.47 | 87.95 ± 9.04 | 91.04 ± 8.31* | 90.80 ± 10.00 | 88.48 ± 12.60 | 88.80 ± 11.62 | 90.13 ± 8.90 | 91.06 ± 9.59* |
| F1 macro | 73.96 ± 10.79 | 70.11 ± 15.54 | 78.70 ± 10.90* | 70.91 ± 19.29 | 72.99 ± 15.65 | 68.25 ± 16.22* | 70.43 ± 18.02 | 76.20 ± 16.36 |

all epochs vs. 11.5% ± 4.9%, p<0.001) and also four times more likely to occur during epochs marked as high confidence (≥80%) by the algorithm (46.4% ± 12.9% of all epochs vs. 11.3% ± 5.7%, p<0.001). Third, there was a significant correlation between the percentage of epochs marked as high confidence by YASA and the percent of epochs with unanimous consensus (*r* = 0.561, p<0.001), meaning that YASA was overall more confident in recordings with higher human interrater agreement.

## Moderator analyses

Having tested the performance of the algorithm on both healthy adults and patients with sleep apnea, we turned our attention to the impact of different moderators on the sleep-staging accuracy. Subsequent analyses are based on the testing set 1 (NSRR), for which – unlike the testing set 2 (DOD) – extensive individual-level demographic and health data was available. As with human sleep scoring, analyses focused on the first factor of age revealed a small but significant negative correlation with accuracy (*r* = −0.128, p=0.002, *Figure 2A*), suggesting a moderate linear decrease of accuracy with age. Second, body composition was not significantly correlated with accuracy (*r* = −0.001, p=0.97, *Figure 2B*). Third, sex was not a significant determinant of accuracy either (Welch's T = −1.635, p=0.103, *Figure 2C*). Fourth, sleep apnea, and specifically the AHI, was negatively correlated with accuracy (*r* = −0.29, p<0.001, *Figure 2E*) as well as the overall confidence level of the algorithm (calculated by averaging the epoch-by-epoch confidence across the entire night, *r* = −0.339, p<0.001). This would indicate that, to a modest degree, the performance and confidence of the algorithm can lessen in those with severe sleep apnea indices, possibly mediated by an increased number of stage transitions in patients with sleep apnea (see above stage transition findings). Consistent with the latter, the percentage of (human-defined) stage transitions was significantly correlated with the AHI (*r* = 0.427, p<0.001). Finally, race was a significant predictor of accuracy (ANOVA, F(3, 581) = 3.281, p=0.021, *Figure 2D*). Pairwise post hoc tests adjusted for multiple comparisons with Tukey's method revealed

that accuracy was lower in Hispanic than in non-Hispanic White individuals (p=0.032). However, this effect may be driven by the imbalance in sample size between these two categories (n = 40 vs. n = 342, respectively). No other pairwise comparison between race categories was significant.

To better understand how these moderators influence accuracy variability, we analyzed the relative contribution of the moderators using a random forest analysis. Specifically, we included all the aforementioned demographics variables in the model, together with medical diagnosis of depression, diabetes, hypertension, and insomnia, and features extracted from the ground-truth sleep scoring such as the percentage of each sleep stage, the duration of the recording, and the percentage of stage transitions in the hypnograms. The outcome variable of the model was the accuracy score of YASA against ground-truth sleep staging, calculated separately for each night. All the nights in the testing set 1 were included, leading to a sample size of 585 unique nights. Results are presented in *Supplementary file 3b*. The percentage of N1 sleep and percentage of stage transitions – both markers of sleep fragmentation – were the two top predictors of accuracy, accounting for 40% of the total relative importance. By contrast, the combined contribution of age, sex, race, and medical diagnosis of insomnia, hypertension, diabetes, and depression accounted for roughly 10% of the total importance.

## Features importance

The 20 most important features of the model are shown in *Figure 1—figure supplement 7*. 11 out of these top 20 features are derived from the electroencephalogram (EEG) signal, 7 from the electrooculogram (EOG), and 1 from the electromyogram (EMG). This indicates that EEG is the most important signal for accurate sleep staging, followed by EOG and to a lesser extent EMG. Time elapsed from the beginning of the recording was also present in the top 20 features, a result consistent with the well-known asymmetry of sleep stages across the night (i.e., more N3 sleep in the first half of the night and more REM sleep in the second half). Consistent with these findings, the algorithm retained high levels of accuracy when using only a single-channel EEG (median accuracy across the 585 nights of the testing set 1 = 85.12%, median kappa = 0.782) or a combination of one EEG and one EOG (median accuracy = 86.92%, median kappa = 0.809). In other words, using only one EEG and one EOG led to an ~0.5% decrease in median accuracy compared to a full model including one EEG, one EOG, one EMG, as well as age and sex.

The single most important feature of the classifier was the absolute power of the EOG power spectrum, followed by the fractal dimension, absolute power, and beta power of the EEG signal. It is relevant that several of these top features are relatively uncommon, nonlinear features (i.e., fractal dimension and permutation entropy). These novel features may capture unique electrophysiological properties of each sleep stage that are missed by traditional linear and/or spectral features. Similarly, 9 out of the 20 most important features include some form of temporal smoothing/normalization (e.g., centered 7.5 min triangular-weighted rolling average). This highlights the importance of taking into account the neighboring epochs – both past and future – for accurate sleep staging (see also *Supplementary file 3a*).

## Software implementation

Implementation of the algorithm is completely open-source and freely available. Our sleep algorithm, colloquially termed YASA (*Vallet, 2018*) (https://github.com/raphaelvallat/yasa), is part of a broader sleep analysis package (or 'library'), written in Python. In addition to the automatic sleep-staging module we describe here, YASA also includes several additional functions such as automatic detection of sleep spindles and slow-waves, automatic artifact rejection, calculation of sleep statistics from a hypnogram, spectral power estimation (e.g., *Figure 3B*), and phase-amplitude coupling. However, use of the basic sleep-staging module is not at all contingent on a desire to quantify any of these metrics. Simply that they are on offer as additional tools in the software suit, should the user wish.

YASA comes with extensive documentation and is released under a BSD-3 Clause license, part of the Open Source Initiative, and can be directly installed with one simple line of code from the Python Package Index repository (in a terminal: pip install yasa). The source code of YASA is freely and publicly hosted on GitHub, and follows standard release cycles with a version-tracking of all the changes made to the code and/or pretrained classifiers. That is, users can choose to ignore the most recent versions and keep a specific version of the code and staging algorithm, which is useful, for

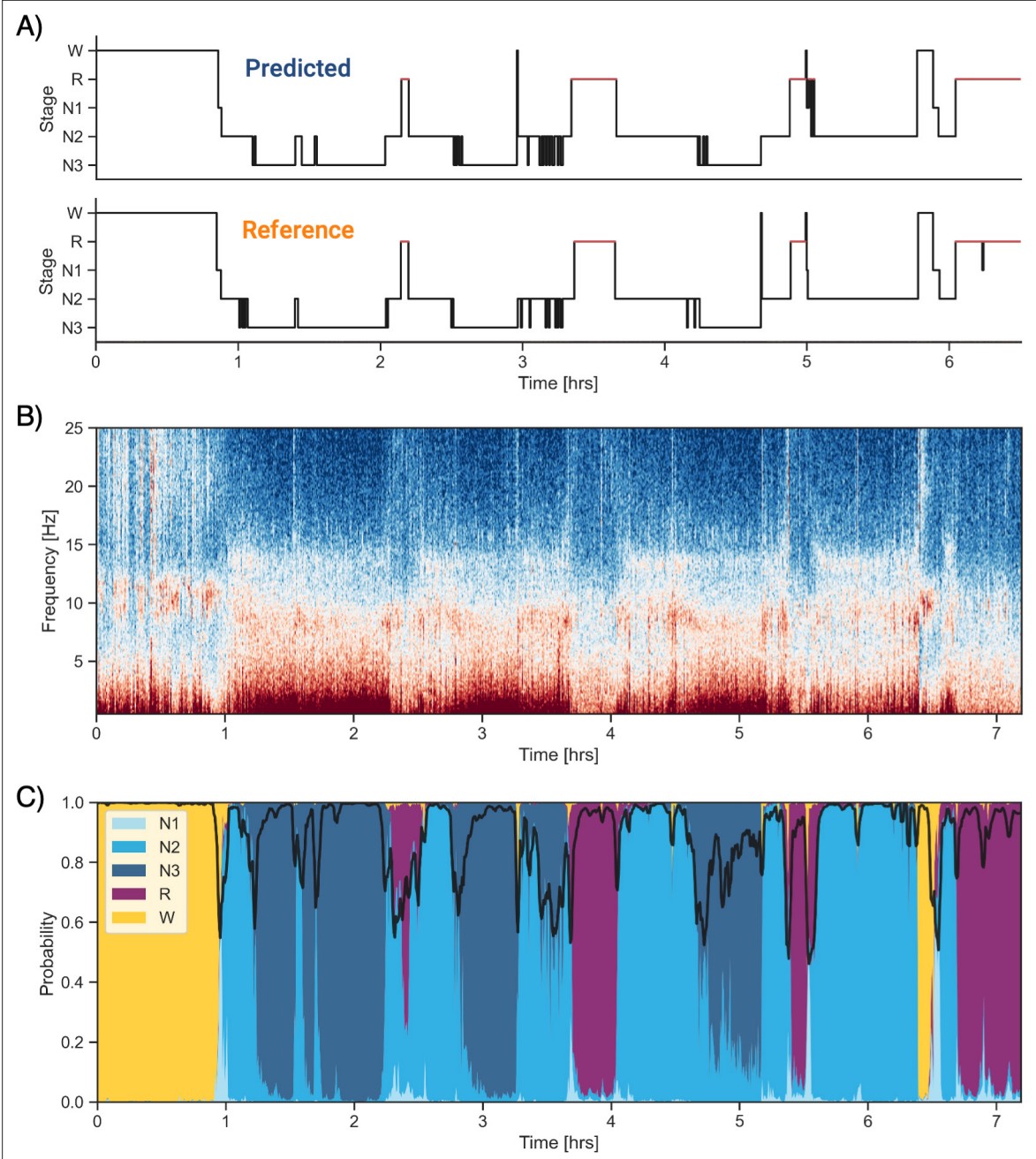

**Figure 3.** Example of data and sleep stages prediction in one subject. (**A**) Predicted and ground-truth ( = human-scored) hypnogram in a healthy young female (CFS dataset, 24 years old, Black, apnea-hypopnea index [AHI] < 1). The agreement between the two scoring is 94.3 %. (**B**) Corresponding full-night spectrogram of the central electroencephalogram (EEG) channel. Warmer colors indicate higher power in these frequencies. This type of plot can be used to easily identify the overall sleep architecture. For example, periods with high power in frequencies below 5 Hz most likely indicate deep non-rapid eye movement (NREM) sleep. (**C**) Algorithm's predicted cumulative probabilities of each sleep stage at each 30 s epoch. The black line indicates the confidence level of the algorithm. Note that all the plots in this figure can be very easily plotted in the software.

The online version of this article includes the following figure supplement(s) for figure 3:

**Figure supplement 1.** Code snippet illustrating the simplest usage of the algorithm.

**Figure supplement 2.** Code snippet illustrating the usage of the algorithm on multiple European Data Format (EDF) files.

example, in longitudinal studies where the preprocessing and analysis steps should stay consistent across time.

The general workflow to perform the automatic sleep staging is described below. In addition, we provide code snippets showing the simplest usage of the algorithm on a single European Data Format (EDF) file (*Figure 3—figure supplement 1*) or on a folder with multiple EDF files (*Figure 3—figure supplement 2*). First, the user loads the PSG data into Python. Assuming that the PSG data are stored in the gold-standard EDF, this can be done in one line of code using the MNE package (*Gramfort et al., 2014*), which has a dedicated function to load EDF files (https://mne.tools/stable/generated/mne.io.read_raw_edf.html). Second, the automatic sleep staging is performed using the algorithm's sleep-staging module (https://raphaelvallat.com/yasa/build/html/generated/yasa.SleepStaging.html). The only requirement is that the user specify the name of the EEG channel they want to apply the detection (preferentially a central derivation such as C4-M1). The algorithm is now ready to stage the data. Should the user wish, there is the option of naming EOG and EMG channels (preferentially a chin EMG). The user can also include ancillary if desired, such as age and sex, which can aid in improving the accuracy of the algorithm, though this is not at all necessary for high accuracy (as described earlier).

Regarding the processing steps of the algorithm, sleep staging is performed with the `predict` function. This function automatically identifies and loads the pretrained classifier corresponding to the combination of sensors/metadata provided by the user. While pretrained classifiers are natively included in the algorithm, a user can define custom pretrained classifiers if they wish, which can be tailored to their specific use cases. Once again, however, this is just for flexibility of use and is not at all required for the algorithm to perform with high accuracy. Parenthetically, this flexibility can also be leveraged for a variety of use cases, for example, scoring rodent sleep data instead of human sleep data.

In addition to stage scoring, implementation of the `predict_proba` function provides the ability for a user to quantify the probability certainty of each stage at each epoch. This probability certainty can then be used to derive a confidence score, should that be desired, although again this is not necessary for standard sleep staging by a user (*Figure 3C*). Finally, the predicted hypnogram and stage probabilities can easily be exported into a text or CSV file using standard Python functions, should the user wish, though once again this is not required.

One limitation of automated data analysis methods is that they are computationally demanding (*Fiorillo et al., 2019*), often requiring specific higher-end computer systems that are costly. To avoid this limitation, the current algorithm is designed to be fast and highly memory-effective by leveraging high-performance tools in Python (e.g., Numpy's vectorization and Numba compiling). As a consequence, a full-night PSG recording sampled at 100 Hz is typically processed in less than 5 s on a rudimentary, basic consumer-level laptop.

## Discussion

We sought to develop a sleep-staging algorithm that (1) matches human-scorer accuracy, (2) has been trained on a large and heterogeneous data set, (3) is easy to implement by most individuals, (4) is computational low-demanding and can therefore be run on a basic laptop, and (5) is entirely free, and thus easily adopted by researchers, clinicians, and commercial ventures.

### Performance

The algorithm displayed high levels of accuracy, matching those observed across human interrater agreement (*Fiorillo et al., 2019*). First, the median agreement between the algorithm and human scoring across 585 testing nights from the NSRR database was 87.5% – and was superior to 90% when only considering epochs that occurred during a stable period of sleep, that is, not around a stage transition. Second, the performance of YASA was tested on a previously unseen dataset of 25 healthy adults and 50 patients with OSA. Each night of this dataset was scored by five registered experts. The median accuracy of the algorithm against the consensus of the five experts was 86.6% for healthy adults and 84.3% in patients. In both samples, the accuracy of YASA was not significantly different from any of the individual human scorers. Furthermore, comparison against two recent deep-learning algorithms (*Stephansen et al., 2018*; *Perslev et al., 2021*) showed that the accuracy of

YASA against consensus scoring was on par (i.e., not statistically different) with the two existing algorithms for healthy adults. However, YASA performed worse than the *Perslev et al., 2021* algorithm in patients with OSA by 2.4% (as discussed in the 'Limitations and future directions' section).

Regarding individual sleep stages, the algorithm showed excellent classification performance for N2 sleep, N3 sleep, REM sleep and wakefulness, and moderate agreement for N1 sleep. These results are consistent with human interrater agreement, which has been found to be consistently lower in N1 sleep compared to the other sleep stages (*Malhotra et al., 2013*; *Norman et al., 2000*; *Rosenberg and Van Hout, 2013*). Furthermore, the algorithm was successful in preserving the overall distribution of sleep stages across the night, such that it is neither over- or underestimating a specific sleep stage.

Beyond a basic sleep-stage classification, an advantage of the algorithm is its ability to provide probability (i.e., likelihood) value for each individual epoch of each stage (*Stephansen et al., 2018*). These probabilities inform the users about the confidence level of the algorithm. As such, the algorithm provides a unique feature addition for sleep scoring that extends beyond sleep-staging quantification and layers atop sleep-staging qualitative assessment. Proving the validity of this measure, the accuracy of the algorithm was statistically superior in epochs that were flagged as high-confidence, reaching ~95% agreement with human scoring. In addition, recordings with a higher average confidence had significantly higher accuracy. Among many other advantages, such an ability could be used in a semi-supervised scoring approach if desired, wherein a human scorer can focus expressly on the selection of low-confidence epochs and/or recordings for attention, thus limiting time investment.

## Generalizability

The power and utility of an algorithm is not just determined by its accuracy, but also by the characteristics of the underlying dataset that was used for training and validation (*Fiorillo et al., 2019*). For example, an algorithm showing excellent performance but that was trained on a very small and homogeneous sample size (e.g., 10–20 healthy young adults from the same demographic group and recorded using the same device) will typically have low utility for other use cases, such as scoring data from another PSG device, or data from patients with sleep disorders. The generalizability of such algorithms can therefore be compromised, with the high performance most likely indicating overfitting to the small training dataset.

Seeking to avoid this issue, our algorithm was trained and evaluated on more than 30,000 hr of PSG human sleep recordings across almost 4000 nights, from numerous independent and heterogeneous datasets. These datasets included participants from a wide age range, different geographical locations, racial groups, sex, body composition, health status, and sleep disorders. Importantly, these independent studies involved different recording devices and settings (e.g., montage, sampling rate). Such a high heterogeneity is paramount to ensure a high reliability and generalizability of the algorithm.

Moderator analyses showed that the performance of the algorithm was unaffected by sex and body composition. Consistent with human scoring (*Muehlroth and Werkle-Bergner, 2020*; *Norman et al., 2000*), the performance of the algorithm was sensitive to advancing age and increasing severity of sleep apnea, although accuracy remained high in patients with sleep apnea (~84% in the DOD-Obstructive validation dataset). The latter two circumstances can be explained by the increase in the number of stage transitions and the increase in the proportion of N1 sleep typically seen in aging and sleep apnea (*Norman et al., 2000*; *Ohayon et al., 2004*). Indeed, the majority of such scoring errors were located around stage transitions and/or in N1 sleep, which is consistent with findings of human interrater error of these same circumstances (*Norman et al., 2000*).

Another problematic challenge in the field of automated sleep staging is the fact that each algorithm is usually tested on a different (often nonpublicly available) dataset, which renders the comparison of algorithms nearly impossible. Adding to this, prior approaches are also heterogeneous in the performance evaluation metrics used to validate the algorithm, as well as in the PSG channel(s) on which the algorithm is applied – with some algorithms using only a single EEG channel and others a multichannel approach (*Fiorillo et al., 2019*). Addressing these issues, the current algorithm was trained and tested on publicly available datasets, using standard classification metrics (e.g., accuracy, F1-scores, confusion matrix) and following recent guidelines for performance evaluation (*Menghini et al., 2021*). Furthermore, the built-in flexibility of the algorithm allows for different combinations of

channels to be used, should a user wish. For all these reasons, we hope that the algorithm is not only of broad utility, but will facilitate replication of our work and comparison to existing and future works.

## Ease-of-use and computationally low demand

To facilitate wide adoption by the sleep community, it is of utmost importance that any algorithm can be used and understood by all parties concerned (e.g., students, researchers, clinicians, technicians), no matter the level of technical expertise. To ensure this, the software has been built with a particular focus on ease-of-use, documentation, and transparency.

First, the end-to-end sleep-staging pipeline can be written in less than 10 lines of Python code, and the software comes with pretrained classifiers that are automatically selected based on the combination of channels used, thus limiting the risk of any error. Second, the software has extensive documentation and includes numerous example datasets, allowing any users to get familiarized with the algorithm before applying it to their own datasets, if they wish (though it is not necessary). Third, the algorithm uses a traditional features-based approach to classify sleep stages instead of a black-box algorithm. These features are described in detail in the documentation and source code of the algorithm, and can be explained to any researchers or clinicians in lay terms. Finally, the sleep staging is done locally on the user's computer, and the data is never uploaded to the cloud or any external servers, thus limiting security and privacy risks (*Fiorillo et al., 2019*), and the need for any connectivity when using the software.

## Free, open source, and community-driven

Another potential reason that automated sleep staging has yet to become a de facto standard is that some algorithms are sitting behind a paywall (e.g., *Malhotra et al., 2013*; *Patanaik et al., 2018*). By contrast, the current algorithm is free and released under a nonrestrictive BSD-3 open-source license. The software, which also includes other sleep analysis tools (e.g., sleep spindle detection, spectral estimation, automatic artifact rejection, phase-amplitude coupling), is hosted on GitHub and has already been downloaded several thousand times at the date of writing (https://static.pepy.tech/badge/yasa). The software can therefore be used by for- or non-profit outlets for free, without constraint.

## Limitations and future directions

Despite its numerous advantages, there are limitations to the algorithm that must be considered. These are discussed below, together with ideas for future improvements of the algorithm. First, while the accuracy of YASA against consensus scoring was not significantly different from the *Stephansen et al., 2018* and *Perslev et al., 2021* algorithms on healthy adults, it was significantly lower than the latter algorithm on patients with OSA. The *Perslev et al., 2021* algorithm used all available EEGs and two (bilateral) EOGs, whereas YASA's scoring was based on one central EEG, one EOG, and one EMG. This suggests that one way to improve performance on this population could be the inclusion of more EEG channels and/or bilateral EOGs. For instance, using the negative product of bilateral EOGs may increase sensitivity to rapid eye movements in REM sleep or slow eye movements in N1 sleep (*Agarwal et al., 2005*; *Stephansen et al., 2018*). Interestingly, the *Perslev et al., 2021* algorithm does not use an EMG channel, which is consistent with our observation of a negligible benefit on accuracy when adding EMG to the model. This may also indicate that while the current set of features implemented in the algorithm performs well for EEG and EOG channels, it does not fully capture the meaningful dynamic information nested within muscle activity during sleep.

Second, an obstacle to the wide adoption of the current algorithm is the absence of a graphical user interface (GUI). Designing and maintaining an open-source cross-platform GUI is in itself a herculean task that requires dedicated funding and software engineers. Rather, significant efforts have been made to facilitate the integration of YASA's outputs to existing sleep GUIs. Specifically, the documentation of the algorithm includes examples on how to load and edit the sleep scores in several free GUIs, such as EDFBrowser, Visbrain, and SleepTrip.

Third, the algorithm was exclusively tested and evaluated on full-night PSG recordings. As such, its performance on shorter recordings, such as daytime naps, is unknown. Further testing of the algorithm is therefore required to validate the algorithm on naps. While YASA can process input data of arbitrary length, performance may be reduced on data that are shorter than the longest smoothing windows used by the algorithm (i.e., shorter than 7.5 min).

Fourth, unlike other deep-learning-based algorithms (e.g., *Perslev et al., 2021*; *Stephansen et al., 2018*), the algorithm does not currently have the ability to score in shorter resolution. We think however that this is not a significant limitation of our work. First, without exception, all of the datasets used either in the training or testing the current algorithm were visually scored by experts using a 30 s resolution. Therefore, even if our algorithm was able to generate sleep scores in a shorter resolution, comparison against the ground-truth would require downsampling the predictions to a 30 s resolution and thus losing the benefit of the shorter resolution. A proper implementation of a shorter resolution scoring requires to train and validate the model on ground-truth sleep scores of the same temporal resolution. Second, the use of 30 s epochs remains the gold standard (as defined in the most recent version of the AASM manual), and as such, the vast majority of sleep centers across the world still rely on 30 s staging. Provided that datasets with sleep scoring at shorter resolution (e.g., 5, 10, 15 s) becomes common, the current algorithm could be modified and retrained to yield such higher-resolution predictions. In that case, the minimum viable resolution would be 5 s, which is the length of the Fast Fourier Transform used to calculate the spectral powers. That said, we firmly believe that a more temporally fine-grained sleep staging will be of benefit to the sleep community in the future. Indeed, such 'high-frequency' staging may improve classification accuracy in error-prone ambiguous epochs (e.g., transition epochs with multiple stages), and has further been shown to better discriminate between healthy and sleep-disordered patients than traditional 30 s-based staging (*Perslev et al., 2021*).

Fifth, the algorithm is not currently able to identify markers of common sleep disorders (such as sleep apnea, leg movements), and as such may not be suited for clinical purposes. It should be noted however that our software does include several other functions to quantify phasic events during sleep (slow-waves, spindles, REMs, artifacts), as well as sleep fragmentation of the hypnogram. Rather than replacing the crucial expertise of clinicians, YASA may thus provide a helpful starting point to accelerate clinical scoring of PSG recordings. Furthermore, future developments of the software should prioritize automated scoring of clinical disorders, particularly apnea-hypopnea events. On the latter, YASA could implement some of the algorithms that have been developed over the last few years to detect apnea-hypopnea events from the ECG or respiratory channels (e.g., *Koley and Dey, 2013*; *Varon et al., 2015*).

A final limitation is that the algorithm is tailored to human scalp data. As such, individuals who may want to use YASA to score intracranial human data, animal data, or even human data from a very specific population will need to adjust the algorithm for their own needs. There are two levels at which the algorithm can be modified. First, an individual may want to retrain the classifier on a specific population without modifying the underlying features. Such flexibility is natively supported by the algorithm and no modifications to the original source code of YASA will be required. However, in some cases, the features may need to be modified as well to capture different aspects and dynamics of the input data (e.g., rodents or human intracranial data). In that case, the users will need to make modifications to the source code of the algorithm, and thus have some knowledge of Python and Git – both of which are now extensively taught in high schools and universities (https://wiki.python.org/moin/SchoolsUsingPython, https://education.github.com/schools). Of note, the entire feature extraction pipeline takes about 100 lines of code and is based on standard scientific Python libraries such as NumPy, SciPy, and Pandas.

## Advantages of YASA against existing tools

Although recent advances in artificial intelligence have made it possible to automate tedious tasks in numerous fields of medicine, the sleep research field still relies on human visual scoring, which is time-consuming and can be prone to subjective bias. It is our opinion that one roadblock that has prevented the wide adoption of automatic sleep staging is the sheer number of algorithms that are published each year, most of which are virtually incomparable to others because they use different testing datasets and/or different evaluation strategies. Here, in an effort of transparency, we have benchmarked our algorithm against two of the most significant algorithms that have been published in recent years (*Perslev et al., 2021*; *Stephansen et al., 2018*), using exactly the same publicly available dataset. As expected, all three algorithms had high levels of accuracy against consensus scoring, that is, on par with typical human interrater agreement. In addition with its high accuracy, there are unique advantages of YASA that should be considered by potential users, which we describe below.

First, and unlike the vast majority of existing algorithms, YASA was not designed for the sole purpose of staging sleep, but rather as an extensive toolbox encompassing the large majority of all analyses used by sleep researchers and clinicians. This includes, among others, (1) the calculation of sleep statistics from the hypnogram, (2) the automatic detection of phasic events during sleep (such as spindles, slow-waves, REMs, and body movements), (3) spectral analysis, as well as (4) more complex and novel analysis methods that we and others have developed, including event-locked cross-frequency coupling (*Helfrich et al., 2018*; *Staresina et al., 2015*) and decomposition of the sleep power spectrum into aperiodic and oscillatory components (*Lendner et al., 2020*; *Wen and Liu, 2016*). As such, our software provides an open-source end-to-end framework for the analysis of PSG sleep data, with sleep staging being a (optional) first step in the analysis pipeline.

A second advantage of YASA compared to existing algorithms is processing speed. Indeed, YASA is orders of magnitude faster than the *Stephansen et al., 2018* algorithm (~10–20 s vs. 10–20 min, including data loading). Noteworthy, while the *Perslev et al., 2021* algorithm is computationally fast, uploading the EDF file to the web server can be slow – in addition to being not adequate for sensitive clinical data. While these differences in processing times may not be problematic for small studies, they can scale dramatically when working with hundreds of recordings.

Third, there are only a limited number of people who have the technical skills and/or required hardware to modify and retrain complex deep-learning neural network architectures, such as the ones implemented in the *Perslev et al., 2021* and *Stephansen et al., 2018* algorithms. In other words, making changes to these algorithms or adapting them to specific needs is practically unfeasible for the average or even advanced user. By contrast, the core YASA algorithm can be easily modified by someone with such rudimentary knowledge of Python, and the full model can be trained on any basic laptop in no more than a couple of hours. Anecdotally, there are a number of ongoing efforts by other research teams to adapt the algorithm to various needs, such as animal data or human intracranial data. It is therefore our hope that such a simple and easily modifiable architecture may foster collaborative development.

## Conclusion

The interest in quantifying and tracking of human sleep has increased exponentially in the past decade (*Fleming et al., 2015*; *Shelgikar et al., 2016*), and demand for a highly automated, accurate, high-speed, and easy-to-implement algorithm for the scoring of human sleep has scaled similarly. Here, we offer an algorithm that seeks to accomplish this need. It is our hope that this free tool has the potential for broad adoption across all outlets (research, clinical, commercial) and use cases. With further validation and feedback from the sleep community, we hope it can become an industry standard method, one that can be built upon, expanded, and refined through cross-collaborative open-source community development.

## Materials and methods
### Datasets

The algorithm was trained on a collection of large-scale independent datasets from the NSRR ( https://sleepdata.org/; *Dean et al., 2016*; *Zhang et al., 2018*) – a web portal funded by the National Heart, Lung and Blood Institute (NHLBI). This database offers access to large collections of deidentified physiological signals and clinical data collected in research cohorts and clinical trials. All data were collected as part of research protocols that were approved by the local institutional review board at each institution, with written and informed consent obtained from each individual before participation. All PSG recordings were scored by trained technicians using standard AASM guidelines (*Iber et al., 2007*; *Silber et al., 2007*). A full description of the datasets can be found on https://sleep-dataorg/. The following datasets were used.

### MESA

The MESA is a multicenter longitudinal investigation of factors associated with the development of subclinical cardiovascular disease and the progression of subclinical to clinical cardiovascular disease in 6814 Black, White, Hispanic, and Chinese-American men and women initially aged 45–84 at baseline in 2000–2002. Full overnight PSG recordings were collected during the last follow-up visit in

2010–2012 and included 2237 participants (age range = 54–95 years). In-home PSG was conducted using the Compumedics Somte System (Compumedics Ltd., Abbotsford, Australia). The recording montage consisted of three cortical EEG (central C4-M1, occipital Oz-Cz, and frontal Fz-Cz leads), bilateral EOG, chin EMG, as well as several other sensors to measure heart rate, respiration, and leg movements. PSG data were sampled at 256 Hz, and a hardware low-pass filter with a cutoff frequency at 100 Hz was applied during recording.

## CFS

The CFS is a family-based study of sleep apnea, consisting of 2284 individuals (46% African-American) from 361 families studied on up to four occasions over a period of 16 years. We used the full overnight PSG recordings from the last exam (visit 5, n = 735, age range = 6–88 years). In-lab PSG was performed using the Compumedics E-Series System. The recording montage included two cortical EEG (C3-Fpz, C4-Fpz), bilateral EOG, and chin EMG. EEG and EOG channels were sampled at 128 Hz, and EMG was sampled at 256 Hz. A hardware bandpass filter with cutoff frequencies at 0.016 Hz and 105 Hz was applied during recording.

## CCSHS

The CCSHS is a population-based pediatric study with objective sleep evaluation, characterized by a large minority representation. We used the most recent visit, which took place between 2006 and 2010 and included in-lab PSG (n = 517, age range = 16–19 years). The PSG recording (Compumedics E-series, Compumedics) included two cortical EEG (C3-Fpz and C4-Fpz), bilateral EOG, and chin EMG. All channels were sampled at 128 Hz, and a hardware high-pass filter with cutoff frequency at 0.15 Hz was applied during recording.

## SHHS

The SHHS is a multicenter cohort study to determine the cardiovascular and other consequences of sleep-disordered breathing. We only included the first visit, which took place between 1995 and 1998 and included full, in-lab PSG from 5,804 participants (age range = 40–89 years). The PSG recording (Compumedics P-Series, Compumedics) included two cortical EEG (C3-M2 and C4-M1), bilateral EOG, and chin EMG. All channels were sampled at 125 Hz, and a hardware high-pass filter with cutoff frequency at 0.15 Hz was applied during recording.

## MrOS

The MrOS is a multicenter observational study of 5994 men, of which the sleep study is a follow-up ancillary study. Overnight PSG recordings were collected for 2907 participants (age range = 65–89 years), which represents about half of the participants included in the parent study. The PSG recordings (Compumedics P-Series, Compumedics) included two cortical EEG (C3-Fpz and C4-Fpz), bilateral EOG, and chin EMG. All channels were sampled at 256 Hz, and a hardware high-pass filter with cutoff frequency at 0.15 Hz was applied during recording.

## CHAT

The CHAT is a multicenter, single-blind, randomized-controlled trial designed to test whether after a 7 -month observation period children, ages 5–9.9 years, with mild to moderate OSA randomized to early adenotonsillectomy will show greater levels of neurocognitive functioning. Overnight PSG recordings were collected between 2007 and 2012 and included 1447 participants (age range = 5–9.9 years), of which 464 were randomized to treatment. The PSG recording consisted of eight cortical EEG (including C3-Fpz and C4-Fpz), bilateral EOG, and chin EMG. All channels were sampled at 200 Hz.

## HomePAP

The HomePAP is a multisite, randomized-controlled trial that enrolled 373 patients (age range = 20–80 years), with a high pretest probability of moderate to severe OSA. We used the in-lab PSG recording, which included six cortical EEG (including C3-Fpz and C4-Fpz), bilateral EOG, and chin EMG. All channels were sampled at 200 Hz.

Each dataset was randomly split into training (up to 600 nights) and testing (up to 100 nights) sets. PSG nights included in the training set were used for model building and training, while PSG nights included in the testing set were used for performance evaluation. Importantly, the training and testing sets were completely separate (i.e., no overlap). The code used to generate the training and testing sets can be found here. Demographics and health data such as the age, sex, race/ethnicity, BMI, AHI (3% desaturation), as well as medical diagnosis of insomnia, depression, diabetes, and hypertension were also provided for each dataset.

To provide an unbiased evaluation of the model on a completely new dataset, we further tested the performance of the algorithm on the DOD (*Guillot et al., 2020*), a publicly available dataset including healthy individuals (DOD-Healthy) and patients with OSA (DOD-Obstructive).

### DOD-Healthy

The *DOD-Healthy* consists of 25 healthy volunteers that were enrolled at the French Armed Forces Biomedical Research Institute's Fatigue and Vigilance Unit in France. Participants were without sleep complaints, aged 18–65, and recruited without regard to gender or ethnicity. PSG recordings were conducted with a Siesta PSG device (Compumedics) and included 12 EEG derivations (C3/M2, F4/M1, F3/F4, F3/M2, F4/O2, F3/O1,FP1/F3, FP1/M2, FP1/O1, FP2/F4, FP2/M1, FP2/O2), 1 EMG and bilateral EOGs, all sampled at 250 Hz.

### DOD-Obstructive

The *DOD-Obstructive* includes 55 patients with clinical suspicion for sleep-related breathing disorder. PSG was conducted at the Stanford Sleep Medicine Center in the US (clinical trial NCT03657329). Individuals clinically diagnosed with sleep disorders other than OSA, suffering from morbid obesity, taking sleep medications, or with certain cardiopulmonary or neurological comorbidities were excluded from the study. PSG recordings were conducted with a Siesta PSG device (Compumedics) and included eight EEG derivations (C3/M2, C4/M1, F3/F4, F3/M2, F4/O2, F3/O1, O1/M2,O2/M1), one EMG and bilateral EOGs, all sampled at 250 Hz.

Individual-level demographics and medical history were not provided for the DOD datasets; group averages for age, BMI, and AHI are reported from *Guillot et al., 2020*. Importantly, no nights from the DOD were used for model training. Each night of the DOD was scored by five clinical experts, thus allowing to compare the performance of the algorithm against a consensus of human scorers (see 'Consensus scoring' section).

### Preprocessing and features extraction

For each PSG night, we extracted a single central EEG, left EOG, and chin EMG. We chose a central EEG (e.g., C4-M1 or C4-Fpz depending on the dataset) since the American Academy of Sleep Medicine (AASM) recommends that a central EEG should be included in all PSG recordings, and it is therefore more likely to be present in a variety of PSG recordings. These signals were then downsampled to 100 Hz to speed up computation time, and bandpass-filtered between 0.40 Hz and 30 Hz. No artifact removal was applied to the PSG data before running the sleep-staging algorithm.

The classification algorithm is based on a machine-learning approach in which a set of 'features' is extracted from the EEG signal, and optionally though non-necessarily, from the EOG and EMG signals as well. Consistent with human sleep staging, features are calculated for each 30 s epoch of raw data. An overview of the features implemented in the algorithm is provided below, focusing on the EEG features (although the EOG and EMG features are virtually identical). All codes used to compute these features are made open-source and freely available to all (see 'Data and code availability' section). The complete list of features included in the final model can be found in *Supplementary file 4*.

These features were selected based on prior work in features-based classification algorithms for automatic sleep staging (*Krakovská and Mezeiová, 2011*; *Lajnef et al., 2015*; *Sun et al., 2017*). For example, it was previously reported that the permutation entropy of the EOG/EMG and the EEG spectral powers in the traditional frequency bands are the most important features for accurate sleep staging (*Lajnef et al., 2015*), thus warranting their inclusion in the current algorithm. Several other features are derived from the authors' previous works with entropy/fractal dimension metrics (*Vallet, 2018*) (https://github.com/raphaelvallat/antropy). Importantly, the features included in the current algorithm were chosen to be robust to different recording montages. As such, we did not include

features that are dependent on the phase of the signal and/or that require specific events detection (e.g., slow-waves, REMs). However, the time-domain features (see below) are dependent upon the amplitude of the signal, and the algorithm may fail if the input data is not expressed in standard units (μV) or has been z-scored prior to applying the automatic sleep staging.

### Time-domain features

The implemented time-domain features comprise standard descriptive statistics, that is, the standard deviation, interquartile range, skewness, and kurtosis of the signal. In addition, several nonlinear features are calculated, including the number of zero-crossings, the Hjorth parameters of mobility and complexity (*Hjorth, 1970*), the permutation entropy (*Bandt and Pompe, 2002*; *Lajnef et al., 2015*), and the fractal dimension (*Esteller et al., 2001*; *Higuchi, 1988*; *Petrosian, 1995*) of the signal.

### Frequency-domain features

Frequency-domains features were calculated from the periodogram of the signal, calculated for each 30 s epoch using Welch's method (*Welch, 1967*) (Hamming window of 5 s with a 50% overlap [ = 0.20 Hz resolution], median-averaging to limit the influence of artifacts). Features included the relative spectral power in specific bands (slow = 0.4–1 Hz, delta = 1–4 Hz, theta = 4–8 Hz, alpha = 8–12 Hz, sigma = 12–16 Hz, beta = 16–30 Hz), the absolute power of the broadband signal, as well as power ratios (delta/theta, delta/sigma, delta/beta, alpha/theta).

### Smoothing and normalization

When scoring sleep, human experts frequently rely on contextual information, such as prior and future epochs around the current epoch being scored (i.e., what was the predominant sleep stage in the last few minutes, what stage is the next epoch). By contrast, feature-based algorithms commonly process one epoch at a time, independently of the past and future epochs, overlooking such contextual temporal information. To overcome this limitation, the current algorithm implemented a smoothing approach across all the aforementioned features. In particular, the features were first duplicated and then smoothed using two different rolling windows: (1) a 7.5 min centered, triangular-weighted rolling average (i.e., 15 epochs centered around the current epoch with the following weights: [0.125, 0.25, 0.375, 0.5, 0.625, 0.75, 0.875, 1., 0.875, 0.75, 0.625, 0.5, 0.375, 0.25, 0.125]), and (2) a rolling average of the last 2 min prior to the current epoch. The optimal time length of these two rolling windows was found using a cross-validation approach (*Supplementary file 3a*).

Critically, there is marked natural interindividual variability in EEG brainwave activity (*Buckelmüller et al., 2006*; *De Gennaro et al., 2008*), meaning that each individual has a unique EEG fingerprint. To take this into account, all the smoothed features were then z-scored across each night, that is, expressed as a deviation from the night's average. The inclusion of such normalized features aids in accommodating the error-potential impact of interindividual variability upon the algorithm, and thus improves final accuracy. The final model includes the 30 s-based features in original units (no smoothing or scaling), as well as the smoothed and normalized version of these raw features. The raw features were included to increase the temporal specificity and keep absolute values that can be compared across individuals regardless of interindividual variability.

Finally, the features set includes time elapsed from the beginning of the night, normalized from 0 to 1. This importantly accounts for the known asymmetry of sleep stages across the night, that is, the predominance of deep NREM sleep in the first half of the night, and conversely, a preponderance of REM and lighter NREM sleep in the second half of the night.

If desired, the user can also add information about the participant's characteristics, such as the age and sex that are known to influence sleep stages (see 'Results'; *Carrier et al., 2001*; *Ohayon et al., 2004*), which the classifier then takes into account during the staging process.

### Machine-learning classification

The full training dataset was then fitted with a LightGBM classifier (*Ke et al., 2017*) – a tree-based gradient-boosting classifier using the following hyper-parameters: 500 estimators, maximum tree depth of 5, maximum number of leaves per tree of 90, and a fraction of 60% of all features selected at random when building each tree. These parameters were chosen to prevent overfitting of the classifier while still maximizing accuracy. Specifically, we performed an automated search of the best

hyper-parameters using a threefold cross-validation on the entire training set. A total of 96 possible combinations of hyper-parameters were tested on the following hyper-parameters: number of estimators, number of leaves, maximum tree depth and feature fraction (which are the most important parameters for controlling overfitting, see the documentation of LightGBM). Importantly, the loss function was defined as

$$\left| acc_{train} - acc_{test} \right| + 4 * \left( 1 - acc_{test} \right)$$

where $acc_{test}$ and $acc_{train}$ are the average cross-validated test/train accuracy, respectively. In other words, the best set of hyper-parameters must maximize the cross-validated accuracy but also minimize the difference in accuracy between the train and test set. To retain optimal performance, the latter was, however, down-weighted by a factor of 4 relative to the former.

In addition, custom sleep-stage weights were passed to the classifier to limit the imbalance in the proportion of sleep stages across the night. Without such weighting, a classifier would favor the most represented sleep stage (N2, ~50% of a typical night), and conversely, would seldom choose the least-represented sleep stage (N1, ~5%). The best weights were found by running a cross-validated parameter search on the full training set, with the average of the accuracy and F1-scores as the optimization metric. A total of 324 possible combinations of class weights were tested. The parameter space was defined as the Cartesian product of N1: [1.6, 1.8, 2, 2.2], N2: [0.8, 0.9, 1], N3/REM/Wake: [1, 1.2, 1.4]. The best weights were 2.2 for N1, 1 for N2 and Wake, and 1.2 for N3 and 1.4 for REM. Python code for the grid search of the best class weights can be found here. The pretrained classifier was then exported as a compressed file (~2 MB) and used to predict (1) a full hypnogram for each night of the testing set and (2) the associated probabilities of each sleep stage for each 30 s epoch.

## Performance evaluation

Performance of the algorithm was evaluated using established standardized guidelines (*Menghini et al., 2021*). First, for each testing night separately, we evaluated the epoch-by-epoch agreement between the human-scoring (considered as the ground-truth) and the algorithm's predictions. Agreement was measured with widely used metrics that included accuracy (i.e., percent of correctly classified epochs), Cohen's kappa (a more robust score that takes into account the possibility of the agreement occurring by chance), and the Matthews correlation coefficient. The latter is thought to be the most robust and informative score for classification as it naturally takes into account imbalance between sleep stages (*Chicco and Jurman, 2020*). For all of the above metrics, higher values indicate higher accuracy agreement. Unless specified, we report the performance of the full model that includes one EEG, one EOG, one EMG, as well as the age and sex of the participant.

In addition, to measure stage-specific performance, we report confusion matrices and F1-scores for each stage separately. The F1-score is defined as the harmonic mean of precision and sensitivity. Put simply, precision is a measure of the *quality* of the prediction (e.g., the proportion of all the epochs classified as REM sleep by the algorithm that were actually labeled as REM sleep in the human scoring), while sensitivity is a measure of the *quantity* of the prediction (e.g., the proportion of all the epochs labeled as REM sleep in the human scoring that were correctly classified as REM sleep by the algorithm). Being the average of both precision and sensitivity, the F1-score is therefore an optimal measure of the algorithm's performance that can be calculated independently for each sleep stage. Higher values indicate superior performance.

Additionally, we conducted discrepancy analysis to test for any systematic bias of the algorithm to over- or underestimate a specific sleep stage. Lastly, we performed moderator analyses to test whether the algorithm maintained a high level of performance across participants of different ages, gender, and health status.

## Consensus scoring

Each night of the DOD was scored by five independent experts. By taking the most voted stage for each epoch, it is therefore possible to build a consensus hypnogram and thus reduce bias caused by low interscorer agreement (*Guillot et al., 2020*; *Perslev et al., 2021*; *Stephansen et al., 2018*). When a tie occurs on a specific epoch, the sleep scoring of the most reliable scorer is used as the reference. The most reliable scorer is the scorer with the highest average agreement with all the other scorers

for the given night. We also compared each human scorer against an unbiased consensus, which was based on the *N-1* remaining scorers (i.e., excluding the current scorer, *Stephansen et al., 2018*). Here again, the most reliable of the *N-1* scorers was used in case of ties.

## Comparison against existing algorithms

We further tested the performance of the current algorithm (YASA) against two recently developed sleep-staging algorithms, namely the *Stephansen et al., 2018* and *Perslev et al., 2021* algorithms. Both algorithms are based on a deep-learning approach and are publicly available on GitHub. The two algorithms were applied on all the nights of the DOD dataset using the same raw data used to test the current algorithm.

Similar to YASA, the *Stephansen et al., 2018* and *Perslev et al., 2021* algorithms did not include any nights of the DOD in their training set, thus ensuring an unbiased comparison. The only difference between the algorithms was in the choice of channels used by the algorithms to predict the sleep stages. While YASA only uses a single EEG (central), one EOG, and one EMG, both the *Stephansen et al., 2018* and *Perslev et al., 2021* algorithms were tested using two EOGs (LOC and ROC) as well as several EEG channels. Specifically, the *Stephansen et al., 2018* algorithm was tested using one chin EMG, two EOGs, and four EEGs (two central and two occipital channels). The *Perslev et al., 2021* algorithm does not require an EMG channel, but uses by default all available EEG channels. For the *Perslev et al., 2021* algorithm, the web portal (https://sleep.ai.ku.dk/) was used to upload each EDF file and download the resulting sleep scores (version: U-Sleep 1.0). For the *Stephansen et al., 2018* algorithm, sleep scores were obtained using the pretrained algorithm on the master branch of the GitHub repository (*Vallet, 2018*) (https://github.com/Stanford-STAGES/stanford-stages; as of July 20, 2021).

For all three algorithms, the ground truth was the consensus scoring of all five human experts. Results were evaluated separately for the healthy individuals (DOD-Healthy) and patients with oOSA (DOD-Obstructive). Statistical comparisons were performed using two-sided paired pairwise t-tests corrected for multiple comparisons using the Holm method.

## Features importance

Features importance was measured on the full training set using the Shapley Additive Explanation algorithm (SHAP; *Lundberg et al., 2020*). Shapley values, which originate from game theory, have several desirable properties that make them optimal for evaluating features contribution in machine learning. Formally, a Shapley value is the average marginal contribution of a feature value across all possible combinations of features. The SHAP contribution of a given feature was computed by first summing the absolute Shapley values on all epochs and then averaging across all sleep stages, thus leading to a single importance score for each feature.

## Data and code availability

All PSG data can be requested from the NSRR website (http://sleepdata.org). The Dreem Open Dataset can be found at https://github.com/Dreem-Organization/dreem-learning-open. The source code and documentation of the algorithm are available at https://github.com/raphaelvallat/yasa. The Python code to reproduce all the results and figures of this paper can be found at https://github.com/raphaelvallat/yasa_classifier. All analyses were conducted in Python 3.8 using scikit-learn 0.24.2 (*Pedregosa et al., 2011*) and lightgbm 3.2.1 (*Ke et al., 2017*).

## Additional information

### Funding
No external funding was received for this work.

### Author contributions
Raphael Vallat, Conceptualization, Formal analysis, Methodology, Software, Writing - original draft, Writing - review and editing; Matthew P Walker, Conceptualization, Formal analysis, Methodology, Writing - original draft, Writing - review and editing

**Author ORCIDs**
Raphael Vallat http://orcid.org/0000-0003-1779-7653
Matthew P Walker http://orcid.org/0000-0002-7839-6389

**Decision letter and Author response**
Decision letter https://doi.org/10.7554/eLife.70092.sa1
Author response https://doi.org/10.7554/eLife.70092.sa2

## Additional files

**Supplementary files**
• Supplementary file 1. Predictions of the automated algorithms for each night of the DOD-Healthy validation dataset. Accuracy refers to the percentage of agreement of the algorithm against the human consensus scoring. Nights are ranked in descending order of agreement between YASA and the consensus scoring.

• Supplementary file 2. Predictions of the automated algorithms for each night of the DOD-Obstructive validation dataset. Accuracy refers to the percentage of agreement of the algorithm against the human consensus scoring. Nights are ranked in descending order of agreement between YASA and the consensus scoring.

• Supplementary file 3. (A) Cross-validation of the best time length for the temporal smoothing of the features. A total of 49 combinations of past and centered rolling windows were tested, defined as the Cartesian product of the following time lengths for the past rolling average: [none, 1 min, 2 min, 3 min, 5 min, 7 min, 9 min] and the centered rolling weighted average: [none, 1.5 min, 2.5 min, 3.5 min, 5.5 min, 7.5 min, 9.5 min], where none indicates that no rolling window was applied. Cross-validation was performed using a threefold validation on the full training set, stratified by nights, such that a polysomnography (PSG) night was either present in the training and validation set, but never in both at the same time. For speed, only 50 trees were used in the classification algorithm. The 'Mean' column is the average of the accuracy and the five F1-scores. Note that the second best-ranked combination (9.5 min centered) has a slightly higher mean score; however, we chose to use a 7.5 min centered window (rank 1) in our final model because it had higher F1-scores for N2, N3, and rapid eye movement (REM) sleep. (B) Contributors of variability in accuracy. Relative importance (%) was estimated with a random forest on n = 585 nights from the testing set 1. The outcome variable of the model was the accuracy score of YASA against ground-truth sleep staging, calculated separately for each night.

• Supplementary file 4. All the features are calculated for each consecutive 30 s epoch across the night, starting from the first sample of the polysomnography recording. Importantly, the algorithm uses three different versions of all time-domain and frequency-domain features: (1) the raw feature, expressed in the original unit of data (e.g., μV for the standard deviation and interquartile range), (2) a smoothed and normalized version of that feature using a 7.5 min triangular-weighted rolling average, and (3) a smoothed and normalized version of that feature using a past 2 min rolling average. Normalization is done after smoothing on a per-night basis with a robust method based on the 5–95% percentiles. Frequency-domain features are based on a Welch's periodogram with a 5 s window (= 0.2 Hz resolution).

• Transparent reporting form

**Data availability**
All polysomnography data can be requested from the NSRR website (http://sleepdata.org). The Dreem Open Dataset can be found at https://github.com/Dreem-Organization/dreem-learning-open. The source code and documentation of the algorithm is available at https://github.com/raphaelvallat/yasa. The Python code to reproduce all the results and figures of this paper can be found at https://github.com/raphaelvallat/yasa_classifier. All analyses were conducted in Python 3.8 using scikit-learn 0.24.2 and lightgbm 3.2.1.

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
