## [Decision Letter]

**Acceptance summary:**

This paper describes an open-source, high accuracy, and easy-to-use toolbox for automatic sleep scoring. The toolbox also includes a set of graphic tools to visualize raw and processed data as well as scoring performance, which will be of great help for the community.

**Decision letter after peer review:**

Thank you for submitting your article "A universal, open-source, high-performance tool for automated sleep staging" for consideration by *eLife*. Your article has been reviewed by 3 peer reviewers, and the evaluation has been overseen by a Reviewing Editor and Christian Büchel as the Senior Editor. The following individual involved in review of your submission has agreed to reveal their identity: Sophie Bagur (Reviewer #1).

Essential revisions:

This study describes an automatic sleep scoring tool, which classifies features extracted from the EEG signal using state-of-the-art machine learning techniques. The algorithm (YASA) was trained and validated on a large dataset including healthy subjects and patients, from various ethnicities and at different age. The algorithm offers a high level of sensitivity, specificity and accuracy matching or sometimes even exceeding that of typical interscorer agreement. While this open-source tool will certainly benefit the field, the three reviewers have raised major concerns that need to be addressed. The essential revision requirements are listed below, please refer to the reviewer's comments for further details.

1) The study does not cite the work of Stephansen et al., (Stephansen et al., Nature Communications 2018, doi: 10.1038/s41467-018-07229-3) in which several interesting approaches were proposed. As the same public data were used, YASA could (and certainly should) be benchmarked against this work. More generally, a short review of the existing tools, their performance and accessibility would be a welcomed addition to the introduction.

2) The performance of the algorithm (YASA) seems to be reduced when inter-human scorer confidence is low. It would be interesting to the reader to understand the underlying reasons: does it happen when states are stable or during transitions? Is there a relationship between YASA and human confusion matrices? How does the performance vary with human confidence during sleep apnea? For misclassified epochs, is the confidence lower in all stages? In the case of an error, is the second-highest probability typically the correct one?

3) The algorithm performs worse on N1 stage, older individuals and patients presenting sleep disorders (sleep fragmentation). It would be helpful to add a section to the manuscript with some considerations on how accuracy could be improved for these different issues. Finally, would it be possible to determine the origin of the accuracy variability? Do low and high accuracy sessions differ in some aspects?

4) As YASA usually performs poorly on fragmented sleep data (as acknowledged by the authors), it would be interesting to present the results of the validation test separately for the healthy volunteers (N-25) and the patients (N=55).

5) Nights were cropped to 15 minutes before and after sleep to remove irrelevant extra periods of wakefulness or artefacts on both ends of the recording. This represents an issue for the computation of important sleep measures such as sleep efficiency and latency as the onset/offset of sleep might be missed.

6) Using 30s epochs seems unnecessary with current computing resource, and this could potentially affect the resolution of the scoring. Along the same lines, using a 5min long smoothing window is not justified and may affect the resolution of transition times, especially during N1. Finally, smoothing should perhaps be assymetrical and restricted to the preceding period, not the following.

7) The study should provide more detailed regarding how the algorithm was prevented from overfitting. Furthermore, it is unclear which features were used exactly whether the features used by the algorithm can be changed by future users, this would be an interesting addition.

8) The algorithm was trained on nighttime sleep data thus cannot be immediately compared to daytime (nap) sleep scoring. It is therefore unjustified to use the word "universal" in the title.

*Reviewer #1 (Recommendations for the authors):*

Details of algorithm predictions:

The authors provide some evidence that the YASA algorithm tends to be mistaken and have low confidence when human inter-scorer variability is high but they only focus on transition periods. The readout of the algorithm's confidence is an interesting tool to assess this. It would therefore be useful for the authors to more systematically explore whether it is related to human scored agreement:

– Correlate epoch by epoch confidence with inter-scorer agreement both for transition and stable periods.

– YASA underperforms on sleep apnea patients, is the confidence also lower and human inter-scorer reliability lower?

– How well do YASA confusion matrices match inter-scorer confusion matrices?

The overall performance of the algorithm is well within the range of the state of the art. It would however be useful to provide some insight into why and how the algorithm fails for those data points with lower accuracy (<70%):

– Does the overall confidence of the algorithm predict the overall accuracy of a given night of sleep? If so this could allow users to exclude untrustworthy nights from further analysis.

– Since accuracy varies between 90 and 65% could the authors show where this variability occurs? (ie general change in accuracy or specific sleep stages more impacted than others).

– It would be useful if the authors could provide as a supplement example predicted and ground truth hypnograms for high, low and median accuracy levels.

Possible algorithm extensions:

The use of 30s epoch is a historical quirk and a way of lightening the load on human scorers but ceases to be relevant for automated scoring, particularly given the problem arising from epochs at transition periods (ex: sleep apnea patients). The possibility of smaller windows ore sliding 30s epochs could potentially refine sleep analysis. Implementation and investigation of this may be beyond the scope of the manuscript but I feel this advantage should be at least discussed.

Methodological details:

The smoothing window used by the authors (5min) seems quite large. Such a long window could be detrimental in cases of fragmented sleep (ex: sleep apnea patients). The authors claim this allows for an optimal compromise between short and long term dynamics. Could the authors back up this claim from the literature or their own model fitting?

The parameters used for the GBM classifier are said to be chose to prevent overfitting. How was this assessed? Did the authors perform a hyperparameter search by splitting the training data set?

The YASA framework allows for artefact removal, was this step performed on the data before running the sleep staging algorithm?

Please provide a definition of the Shapley value in the methods section.

*Reviewer #2 (Recommendations for the authors):*

1. As mentioned in the previous section, I am very surprised that the work of Stephansen et al., (Stephansen et al., Nature Communications 2018, doi: 10.1038/s41467-018-07229-3) is not cited. It is extremely relevant as Stephansen et al., also used a large amount of public data to train and test an automated sleep scoring algorithm (based on neural networks). They are very similar ideas developed in this article (for example they also introduced the use of probabilistic scoring with hypnodensities). Some ideas could also be worth implementing with the proposed algorithm such as testing the impact of common sleep disturbances or shortening the epoch window. Finally, most if not all the data used in Stephansen et al., are available on the SleepData.org platform (used by the authors) so the authors could directly benchmark the performance of the two approaches. From my personal experience, Stephansen algorithm has three disadvantages compared to the present algorithm: (i) it is slower and heavier, (ii) it requires python machine-learning packages such as TensorFlow, which are frequently updated, (iii) it is very difficult to modify and understand how it works (black box). I think it would be interesting for the authors to show how their approach could mitigate these issues.

2. The tool would greatly benefit from a graphical interface to install and use the software. Another interesting function could be to process a list of EDF files (for example, all the EDF files in a given folder). From personal experience again, an automated sleep scoring algorithm will not be used in clinical settings at a large scale without a GUI.

3. I think the algorithm will be closer to the current consensus if the smoothing was restricted to preceding and not following epochs. Also, could the authors examine the impact of the duration of the smoothing period on performance? A too large smoothing might decrease the performance for transitory stages such as N1.

4. Since the proposed algorithms can provide a probabilistic score, it would be interesting to explore further the differences between the human gold-standard scores and the algorithm. For example, for misclassified epochs, is the confidence lower in all stages? In the case of an error, is the second-highest probability typically the correct one? Related to the same point, when writing "for epochs ﬂagged by the algorithm as high-conﬁdence ({greater than or equal to}80% conﬁdence) than in epochs with a conﬁdence below 80%", can the authors indicate the % of epochs {greater than or equal to} and < to 80% confidence? Could the authors show the distribution of these confidences per sleep stage?

5. In the "Descriptive Statistics" section, could the authors add a table assessing the difference between the training and test sets for the variables mentioned in the text?

6. How was the AHI obtained? Was it provided for each dataset and individual or computed by the authors?

7. "These results match recently developed deep-learning-based algorithms for automatic sleep staging (Guillot et al., 2020; Perslev et al., 2021)." Citing the Stephansen paper is important here, especially since the authors could apply their algorithms to the same data. Also, the Stephansen studies should be cited when discussing the advantage of a probabilistic scoring (they coined the term hypnodensity).

8. Out of curiosity, could the list of features used by the algorithms be changed by future users? This could be an interesting feature for future developments.

*Reviewer #3 (Recommendations for the authors):*

In this study, Vallat and Walker describe a new sleep scoring tool that is based on a classification algorithm using machine-learning approaches in which a set of features is extracted from the EEG signal. The algorithm was trained and validated on a very large number of nocturnal sleep datasets including participants with various ethnicities, age and health status. Results show that the algorithm offers a high level of sensitivity, specificity and accuracy matching or sometimes even exceeding that of typical interscorer agreement. Importantly, a measure of the algorithm's confidence is provided for each scored epoch in order to guide users during their review of the output. The software is described as easy to use, computationally low-demanding, open source and free.

This paper addresses an important need in the field of sleep research. There is indeed a lack of accurate, flexible and open source sleep scoring tools. I would like to commend the authors for their efforts in providing such a tool for the community and for their adherence to the open science framework as the data and codes related to the current manuscript are made available. I predict that this automated tool will be of use for a large number of researchers in the field. I also enjoyed reading the paper that is nicely written. However, I have some concerns listed below that need to be addressed and I also provided comments that might help improving the overall quality of the paper.

– There are some overstatements that need to be toned down.

– In the title, the word "universal" should be removed. The tool was trained and validated on nocturnal sleep data. Sleep characteristics (eg duration and distribution of sleep stages etc.) are different, for example, during diurnal sleep (nap) and the algorithm might not perform as well on nap data. Note that there is a large number of scientific studies in which diurnal, instead of nocturnal, sleep paradigms are used as these paradigms are easier to implement in lab settings. The algorithm being optimized for nocturnal sleep (with eg the definition of specific sleep stage weights that are specific to nocturnal recordings), it is unknown how it would perform on nap data for example.

– In the abstract, the following sentence needs to be altered: "This tool offers high sleep-staging accuracy matching or exceeding human accuracy". Exceeding human accuracy might be misleading as human scores are used as the ground-truth in the validation process. The algorithm exceeds the accuracy of some human scorers and matches the scores of the best scorer.

– Acknowledgement of – and comparisons to – already available tools in the field

There are plenty of automated sleep scoring tools available in the field (most of them are not open source and rather expensive though – as noted by the authors). A short review of the existing tools, their performance and accessibility would be a welcomed addition to the introduction. In line with this comment, other published algorithms were tested on the same set of data used to train and validate the present algorithm. The authors refer to their work for comparison between algorithms but it would be a very nice addition to the paper to provide (and test for) such comparisons. It is currently unclear whether the present algorithm performs any better than algorithms already available in the field.

– Further improvements

The algorithm performs worse on N1 stage, older individuals and patients presenting sleep disorders (sleep fragmentation). It would be helpful to add a section to the manuscript with some considerations on how accuracy could be improved for these different issues. Eg, should one consider training the algorithm on older datasets in order to improve accuracy of scoring when studying aging? Same applies to sleep disorders. It is currently unclear whether the variety of datasets used to train the algorithm is beneficial in these cases.

In the same vein, as algorithms usually perform poorly on fragmented sleep data (as acknowledged by the authors), it would be interesting to present the results of the validation test separately for the healthy volunteers (N-25) and the patients (N=55).

– Justification for some methodological choices

– Nights were cropped to 15 minutes before and after sleep to remove irrelevant extra periods of wakefulness or artefacts on both ends of the recording. This represents an issue for the computation of important sleep measures such as sleep efficiency and latency as the onset/offset of sleep might be missed.

– How were features selected? A description of the features needs to be provided. Which features were Z scored exactly and why not all?

– The custom sleep stage weights procedure is unclear. Why was only a sub-sample used to determine best weights?

– Was the smoothing done with 5min or 15min rolling average? While it is critical to include this temporal window, it looks rather wide. Human scorers usually don't go that far back when scoring but usually just a few epochs (ie 2-3 which is 1min30). What was the rational for choosing such a wide temporal window?

– It is currently unclear when / how the EEG and EMG data were analyzed.

---

## [Author Response]

Essential revisions:This study describes an automatic sleep scoring tool, which classifies features extracted from the EEG signal using state-of-the-art machine learning techniques. The algorithm (YASA) was trained and validated on a large dataset including healthy subjects and patients, from various ethnicities and at different age. The algorithm offers a high level of sensitivity, specificity and accuracy matching or sometimes even exceeding that of typical interscorer agreement. While this open-source tool will certainly benefit the field, the three reviewers have raised major concerns that need to be addressed. The essential revision requirements are listed below, please refer to the reviewer's comments for further details.1) The study does not cite the work of Stephansen et al., (Stephansen et al., Nature Communications 2018, doi: 10.1038/s41467-018-07229-3) in which several interesting approaches were proposed. As the same public data were used, YASA could (and certainly should) be benchmarked against this work. More generally, a short review of the existing tools, their performance and accessibility would be a welcomed addition to the introduction.

Thanks so much for pointing this out. We have now added this relevant reference throughout the manuscript. To build on the reviewer’s point, the current algorithm and Stephansen’s algorithm did not use the same public data. The Stephansen 2018 algorithm was trained and validated on “10 different cohorts recorded at 12 sleep centers across 3 continents: SSC, WSC, IS-RC, JCTS, KHC1, AHC, IHC, DHC, FHC and CNC”, none of which are included in the training/testing sets of the current algorithm. Nevertheless, we certainly agree that the manuscript will benefit from a more extensive comparison against existing tools. To this end, we have made several major modifications to the manuscript.

First, we have added a dedicated paragraph in the Introduction to review existing sleep staging algorithms:

“Advances in machine-learning have led efforts to classify sleep with automated systems. Indeed, recent years have seen the emergence of several automatic sleep staging algorithms. While an exhaustive review of the existing sleep staging algorithms is out of the scope of this article, we review below — in chronological order — some of the most significant algorithms of the last five years. For a more in-depth review, we refer the reader to Fiorillo et al., 2019. The Sun et al., 2017 algorithm was trained on 2,000 PSG recordings from a single sleep clinic. The overall Cohen's kappa on the testing set was 0.68 (n=1,000 PSG nights). The “Z3Score” algorithm (Patanaik et al., 2018) was trained and evaluated on ~1,700 PSG recordings from four datasets, with an overall accuracy ranging from 89.8% in healthy adults/adolescents to 72.1% in patients with Parkinson’s disease. The freely available “Stanford-stage” algorithm (Stephansen et al., 2018) was trained and evaluated on 10 clinical cohorts (~3,000 recordings). The overall accuracy was 87% against the consensus scoring of several human experts in an independent testing set. The “SeqSleepNet” algorithm (Phan et al., 2019) was trained and tested using a 20-fold cross-validation on 200 nights (overall accuracy = 87.1%). Finally, the recent U-Sleep algorithm (Perslev et al., 2021) was trained and evaluated on PSG recordings from 15,660 participants of 16 clinical studies. While the overall accuracy was not reported, the mean F1-score against the consensus scoring of five human experts was 0.79 for healthy adults and 0.76 for patients with sleep apnea.”

Second, and importantly, we now perform an in-depth comparison of YASA’s performance against the Stephansen 2018 algorithm and the Perslev 2021 algorithm using the same data for all three datasets. Specifically, we have applied the three algorithms to each night of the Dreem Open Datasets (DOD) and compared their performance in dedicated tables in the Results section (Table 2 and Table 3). This procedure is fully described in a new “Comparison against existing algorithms” subsection of the Methods. None of these algorithms included nights from the DOD in their training set, thus ensuring a fair comparison of the three algorithms. Related to point 4 of the Essential Revisions, performance of the three algorithms are reported separately for healthy individuals (DOD-Healthy, n=25) and patients with sleep apnea (DOD-Obstructive, n=50). To facilitate future validation of our algorithm, we also provide the predicted hypnograms of each night in Supplementary File 1 (healthy) and Supplementary File 2 (patients).

Overall, the comparison results show that YASA’s accuracy is not significantly different from the Stephansen 2018 algorithm for both healthy adults and patients with obstructive sleep apnea. The accuracy of the Perslev 2021 algorithm is not significantly different from YASA in healthy adults, but is higher in patients with sleep apnea. However, it should be noted that while the YASA algorithm only uses one central EEG, one EOG and one EMG, the Perslev 2021 algorithm uses all available EEGs as well as two EOGs. This suggests that adding more EEG channels and/or using the two EOGs may improve the performance of YASA in patients with sleep apnea. Though an important counterpoint is that YASA requires a far less extensive array of data (channels) to accomplish very similar levels of accuracy, which has the favorable benefit of reducing analysis computational and processing demands, improves speed of analysis (i.e. a few seconds per recording versus ~10 min for the Stephansen 2018 algorithm), and is amenable to more data recordings since many may not have sufficient EEG channels. All these points are now discussed in detail in the new “Limitations and future directions” subsection of the Discussion (see point 3 of the Essential Revisions).

As an example of these incorporated changes include new Table 2, which shows the comparison of YASA against the consensus scoring, each of the individual human scorers as well as the Stephansen 2018 and Perslev 2018 algorithms.

2) The performance of the algorithm (YASA) seems to be reduced when inter-human scorer confidence is low. It would be interesting to the reader to understand the underlying reasons: does it happen when states are stable or during transitions? Is there a relationship between YASA and human confusion matrices? How does the performance vary with human confidence during sleep apnea? For misclassified epochs, is the confidence lower in all stages? In the case of an error, is the second-highest probability typically the correct one?

We have now conducted additional analyses to examine the relationship between YASA and the human inter-rater agreement, as advised by the reviewer. Building on that analysis, we have revised the Results sections in several ways. First, for the “Testing set 2” subsection of the Results, we now state:

“Additional analyses on the combined dataset (n=75 nights, healthy and patients combined) revealed three key findings. First, the accuracy of YASA against the consensus scoring was significantly higher during stable epochs compared to transition epochs (mean ± STD: 91.2 ± 7.8 vs 68.95 ± 7.7, p<0.001), or in epochs that were marked as high confidence by the algorithm (93.9 ± 6.0 vs 64.1 ± 7.0 for low confidence epochs, p<0.001). Second, epochs with unanimous consensus from the five human experts were four times more likely to occur during stable epochs than transition epochs (46.2 ± 12.1 percent of all epochs vs 11.5 ± 4.9%, p<0.001) and also four times more likely to occur during epochs marked as high confidence (≥80%) by the algorithm (46.4 ± 12.9 percent of all epochs vs 11.3 ± 5.7%, p<0.001). Third, there was a significant correlation between the percentage of epochs marked as high confidence by YASA and the percent of epochs with unanimous consensus (r=0.561, p<0.001), meaning that YASA was overall more confident in recordings with higher human inter-rater agreement.”

For the “Testing set 1” subsection of the Results, we have further added the following:

“The median confidence of the algorithm across all the testing nights was 85.79%. Nights with a higher average confidence had significantly higher accuracy (Figure 1B, r=0.76, p<0.001). […] In addition, when the algorithm misclassified an epoch, the second highest-probability stage predicted by the algorithm was the correct one in 76.3 ± 12.6 percent of the time.”

In addition, we have revised the “Moderator analyses” subsection:

“Fourth, sleep apnea, and specifically the apnea-hypopnea index (AHI) was negatively correlated with accuracy (r=-0.29, p<0.001, Figure 2E), as well as the overall confidence level of the algorithm (calculated by averaging the epoch-by-epoch confidence across the entire night, r=-0.339, p<0.001). This would indicate that, to a modest degree, the performance and confidence of the algorithm can lessen in those with severe sleep apnea indices, possibly mediated by an increased number of stage transitions in patients with sleep apnea (see above stage transition findings). Consistent with the latter, the percentage of (human-defined) stage transitions was significantly correlated with the AHI (r=0.427, p<0.001).”

Finally, the confusion matrices of each individual human scorer and each algorithm can now be found in Supplementary Materials (Figure 1—figure supplement 3-6).

Confusion matrices are reported separately for healthy adults and patients with sleep apnea.

3) The algorithm performs worse on N1 stage, older individuals and patients presenting sleep disorders (sleep fragmentation). It would be helpful to add a section to the manuscript with some considerations on how accuracy could be improved for these different issues. Finally, would it be possible to determine the origin of the accuracy variability? Do low and high accuracy sessions differ in some aspects?

We have addressed these helpful questions with two key revisions to the manuscript. First, we have now added a “Limitations and Future Directions” subsection in the Discussion to present ideas for improving the algorithm, with a particular focus on fragmented nights and/or nights from patients with sleep disorders:

“Despite its numerous advantages, there are limitations to the algorithm that must be considered. These are discussed below, together with ideas for future improvements of the algorithm. First, while the accuracy of YASA against consensus scoring was not significantly different from the Stephansen 2018 and Perslev 2021 algorithms on healthy adults, it was significantly lower than the latter algorithm on patients with obstructive sleep apnea. The Perslev 2021 algorithm used all available EEGs and two (bilateral) EOGs, whereas YASA’s scoring was based on one central EEG, one EOG and one EMG. This suggests that one way to improve performance in this population could be the inclusion of more EEG channels and/or bilateral EOGs. For instance, using the negative product of bilateral EOGs may increase sensitivity to rapid eye movements in REM sleep or slow eye movements in N1 sleep (Stephansen et al., 2018; Agarwal et al., 2005). Interestingly, the Perslev 2021 algorithm does not use an EMG channel, which is consistent with our observation of a negligible benefit on accuracy when adding EMG to the model. This may also indicate that while the current set of features implemented in the algorithm performs well for EEG and EOG channels, it does not fully capture the meaningful dynamic information nested within muscle activity during sleep.”

Second, we have now conducted a random forest analysis to identify the main contributors of accuracy variability. The analysis is described in detail in the “Moderator Analyses” subsection of the Results as well as Supplementary File 3b, the revision now states:

“To better understand how these moderators influence variability in accuracy, we quantified the relative contribution of the moderators using a random forest analysis. Specifically, we included all aforementioned demographics variables in the model, together with medical diagnosis of depression, diabetes, hypertension and insomnia, and features extracted from the ground-truth sleep scoring such as the percentage of each sleep stage, the duration of the recording and the percentage of stage transitions in the hypnograms. The outcome variable of the model was the accuracy score of YASA against ground-truth sleep staging, calculated separately for each night. All the nights in the testing set 1 were included, leading to a sample size of 585 unique nights. Results are presented in Supplementary File 3b. The percentage of N1 sleep and percentage of stage transitions — both markers of sleep fragmentation — were the two top predictors of accuracy, accounting for 40% of the total relative importance. By contrast, the combined contribution of age, sex, race and medical diagnosis of insomnia, hypertension, diabete and depression accounted for roughly 10% of the total importance.”

4) As YASA usually performs poorly on fragmented sleep data (as acknowledged by the authors), it would be interesting to present the results of the validation test separately for the healthy volunteers (N-25) and the patients (N=55).

As requested by the reviewer, we now analyze and report the performance of YASA on the DOD testing set separately for healthy individuals (DOD-healthy) and patients with obstructive sleep apnea (DOD-Obstructive), which can be found in section “Testing set 2”

5) Nights were cropped to 15 minutes before and after sleep to remove irrelevant extra periods of wakefulness or artefacts on both ends of the recording. This represents an issue for the computation of important sleep measures such as sleep efficiency and latency as the onset/offset of sleep might be missed.

This truncation step has now been removed from the pipeline and all the results have been updated accordingly. In addition, we have also removed all other exclusion criteria (e.g. PSG data quality, recording duration, etc) to improve the generalization power of the algorithm, thanks to the suggestions of the reviewer.

6) Using 30s epochs seems unnecessary with current computing resource, and this could potentially affect the resolution of the scoring. Along the same lines, using a 5min long smoothing window is not justified and may affect the resolution of transition times, especially during N1. Finally, smoothing should perhaps be assymetrical and restricted to the preceding period, not the following.

Scoring resolution

We have now added the following paragraph in the “Limitations and Future Directions” subsection of the Discussion:

“Unlike other deep-learning based algorithms (e.g. Stephansen et al., 2018; Perslev et al., 2021), the algorithm does not currently have the ability to score in shorter resolution. We think however that this is not a significant limitation of our work. First, without exception, all of the datasets used either in the training or testing the current algorithm were visually scored by experts using a 30-seconds resolution. Therefore, even if our algorithm was able to generate sleep scores in a shorter resolution, comparison against the ground-truth would require downsampling the predictions to a 30-sec resolution and thus losing the benefit of the shorter resolution. A proper implementation of a shorter resolution scoring requires to train and validate the model on ground-truth sleep scores of the same temporal resolution. Second, the use of 30-sec epochs remains the gold-standard (as defined in the most recent version of the AASM manual) and as such, the vast majority of sleep centers across the world still rely on 30-sec staging. Provided that datasets with sleep scoring at shorter resolution (e.g. 5, 10, 15 seconds) becomes common, the current algorithm could be re-trained to yield such higher-resolution predictions. In that case, the minimum viable resolution would be 5-seconds, which is the length of the Fast Fourier Transform used to calculate the spectral powers.”

Temporal smoothing

We have also conducted a new analysis of the influence of the temporal smoothing on the performance. The results are described in Supplementary File 3a. Briefly, using a cross-validation approach, we have tested a total of 49 combinations of time lengths for the past and centered smoothing windows. Results demonstrated that the best performance is obtained when using a 2 min past rolling average in combination with a 7.5 minutes centered, triangular-weighted rolling average. Removing the centered rolling average resulted in poorer performance, suggesting that there is an added benefit of incorporating data from both before and after the current epoch. Removing both the past and centered rolling averages resulted in the worst performance (-3.6% decrease in F1-macro). Therefore, the new version of the manuscript and algorithm now uses a 2 min past and 7.5 min centered rolling averages. All the results in the manuscript have been updated accordingly. We have now edited the “Smoothing and normalization” subsection of the Methods section as follow:

“In particular, the features were first duplicated and then smoothed using two different rolling windows: (1) a 7.5 minutes centered, and triangular-weighted rolling average (i.e. 15 epochs centered around the current epoch with the following weights: [0.125, 0.25, 0.375, 0.5, 0.625, 0.75, 0.875, 1., 0.875, 0.75, 0.625, 0.5, 0.375, 0.25, 0.125]), and (2) a rolling average of the last 2 minutes prior to the current epoch. The optimal time length of these two rolling windows was found using a parameter search with cross-validation (Supplementary File 3a). […] The final model includes the 30-sec based features in original units (no smoothing or scaling), as well as the smoothed and normalized version of these raw features.”

7) The study should provide more detailed regarding how the algorithm was prevented from overfitting.

We have now edited the “Machine-learning classification” subsection of the Methods as follow:

“The full training dataset was then fitted with a LightGBM classifier (Ke et al., 2017) — a tree-based gradient-boosting classifier using the following hyper-parameters: 500 estimators, maximum tree depth of 5, maximum number of leaves per tree of 90 and a fraction of 60% of all features selected at random when building each tree. These parameters were chosen to prevent overfitting of the classifier while still maximizing accuracy. Specifically, we performed an automated search of the best hyper-parameters using a 3-fold cross-validation on the entire training set. A total of 96 possible combinations of hyper-parameters were tested on the following hyper-parameters: number of estimators, number of leaves, maximum tree depth and feature fraction (which are the most important parameters for controlling overfitting, see the documentation of LightGBM). Importantly, the loss function was defined as:

|acctrain−acctest|+4*(1−acctest)where acctest and acctrain are the average cross-validated test / train accuracy, respectively. In other words, the best set of hyper-parameters must maximize the cross-validated accuracy but also minimize the difference in accuracy between the train and test set. To retain optimal performance, the latter was, however, down-weighted by a factor of 4 relative to the former.”

Furthermore, it is unclear which features were used exactly whether the features used by the algorithm can be changed by future users, this would be an interesting addition.

We have edited the “Preprocessing and features extraction” subsection of the Methods to clarify which features were used, and also make clear the fact that the algorithm includes *both* the raw features (no smoothing, no scaling) and the smoothed-normalized version of these features:

“All code used to compute these features is made open-source and freely available to all (see Data and code availability). The complete list of features included in the final model can be found here. […] The final model includes the 30-sec based features in original units (no smoothing or scaling), as well as the smoothed and normalized version of these raw features.”

Furthermore, we have added a dedicated paragraph in the “Limitations and future directions” subsection of the Discussion to explain how and when the features should be changed by future users:

“A final limitation is that the algorithm is tailored to human scalp data. As such, individuals that may want to use YASA to score intracranial human data, animal data or even human data from a very specific population will need to adjust the algorithm for their own needs. There are two levels at which the algorithm can be modified. First, an individual may want to re-train the classifier on a specific population without modifying the underlying features. Such flexibility is natively supported by the algorithm and no modifications to the original source code of YASA will be required. Second, in some cases (e.g. rodents data), the features may need to be modified as well to capture different aspects and dynamics of the input data (e.g. rodents or human intracranial data). In that case, the users will need to make modifications to the source code of the algorithm, and thus have some knowledge of Python and Git — both of which are now extensively teached in high schools and universities^8,9^. Of note, the entire feature extraction pipeline takes only ~100 lines of code and is based on standard scientific Python libraries such as NumPy, SciPy and Pandas.”

^8^ https://wiki.python.org/moin/SchoolsUsingPython

^9^ https://education.github.com/schools

8) The algorithm was trained on nighttime sleep data thus cannot be immediately compared to daytime (nap) sleep scoring. It is therefore unjustified to use the word "universal" in the title.

Thanks for this thoughtful point. We have now removed the word “universal” from the title. Furthermore, we now discuss the fact that the algorithm has not been validated on naps data in the “Limitations and Future Directions” of the Discussion:

“Third, the algorithm was exclusively tested and evaluated on full nights PSG recordings, and as such its performance on shorter recordings such as daytime naps is unknown. Further testing of the algorithm is therefore required to validate the algorithm on naps. While YASA can process input data of arbitrary length, performance may be reduced on data that are shorter than the longest smoothing windows used by the algorithm (i.e. 7.5 minutes).”

Reviewer #1 (Recommendations for the authors):Details of algorithm predictions:The authors provide some evidence that the YASA algorithm tends to be mistaken and have low confidence when human inter-scorer variability is high but they only focus on transition periods. The readout of the algorithm's confidence is an interesting tool to assess this. It would therefore be useful for the authors to more systematically explore whether it is related to human scored agreement:– Correlate epoch by epoch confidence with inter-scorer agreement both for transition and stable periods.– YASA underperforms on sleep apnea patients, is the confidence also lower and human inter-scorer reliability lower?– How well do YASA confusion matrices match inter-scorer confusion matrices?

We have now conducted several additional analyses to examine the relationship between human inter-rater agreement and YASA’s confidence levels. These are described in full in point 2 of the Essential Revisions. Briefly, our results show that:

1. On the DOD testing set, epochs with unanimous human consensus are four times more likely to occur during stable periods and/or during epochs flagged as high-confidence by the algorithm.

2. On the DOD testing set, YASA is more confident in nights with a higher average human inter-rater agreement (correlation r = 0.56, p<0.001).

3. On the NSRR testing set (which includes the AHI score for each night), the overall confidence level of the algorithm declines with increasing AHI (r=-0.339, p<0.001).

Furthermore, as discussed in point 1 of the Essential Revisions, we now report pairwise comparisons of YASA’s performance against each individual human scorer, for both overall accuracy and stage-specific F1-scores. In addition, we have added the confusion matrices of YASA and each individual human scorer to the Supplementary Materials, separately for healthy individuals and patients with obstructive sleep apnea (Figure 1—figure supplement 3-6).

The overall performance of the algorithm is well within the range of the state of the art. It would however be useful to provide some insight into why and how the algorithm fails for those data points with lower accuracy (<70%):– Does the overall confidence of the algorithm predict the overall accuracy of a given night of sleep? If so this could allow users to exclude untrustworthy nights from further analysis.

The overall confidence of the algorithm for a given night does indeed predict the accuracy for that same night. As described in the Results section:

“The median confidence of the algorithm across all the testing nights was 85.79%. Nights with a higher average confidence had significantly higher accuracy (Figure 1B, r=0.76, p<0.001).”

– Since accuracy varies between 90 and 65% could the authors show where this variability occurs? (ie general change in accuracy or specific sleep stages more impacted than others).

This has been addressed in point 3 of the Essentials Revisions (random forest analysis of the contributors of accuracy variability).

– It would be useful if the authors could provide as a supplement example predicted and ground truth hypnograms for high, low and median accuracy levels.

We have now added two supplementary files with the predicted hypnograms for each night of the DOD-Healthy and DOD-Obstructive datasets. In addition with YASA’s predicted hypnograms, the supplementary files also include the ground-truth hypnogram (consensus scoring) as well as the Stephansen 2018 and Perslev 2021 hypnograms. Nights in the PDF files are ranked in descending order of agreement between YASA and the consensus scoring. Author response image 1 shows an example of a single night in the PDF files.

**Author response image 1. sa2fig1:** 

Possible algorithm extensions:The use of 30s epoch is a historical quirk and a way of lightening the load on human scorers but ceases to be relevant for automated scoring, particularly given the problem arising from epochs at transition periods (ex: sleep apnea patients). The possibility of smaller windows ore sliding 30s epochs could potentially refine sleep analysis. Implementation and investigation of this may be beyond the scope of the manuscript but I feel this advantage should be at least discussed.

Please refer to point 6 of the Essential Revisions (“Scoring Resolution” subsection).

Methodological details:The smoothing window used by the authors (5min) seems quite large. Such a long window could be detrimental in cases of fragmented sleep (ex: sleep apnea patients). The authors claim this allows for an optimal compromise between short and long term dynamics. Could the authors back up this claim from the literature or their own model fitting?

This has been addressed in point 6 of the Essential Revisions (“Temporal Smoothing” subsection).

The parameters used for the GBM classifier are said to be chose to prevent overfitting. How was this assessed? Did the authors perform a hyperparameter search by splitting the training data set?

Thanks for this. We refer the reviewer to point 7 of the Essential Revisions where we address just this important issue.

The YASA framework allows for artefact removal, was this step performed on the data before running the sleep staging algorithm?

We did not apply artefact removal prior to running the algorithm. This has been clarified in the Methods section:

“No artefact removal was applied to the PSG data before running the sleep staging algorithm.”

Please provide a definition of the Shapley value in the methods section.

We have now added a dedicated “Features Importance” section in the Methods:

“Feature importance was measured on the full training set using the Shapley Additive Explanation algorithm (SHAP; Lundberg et al., 2020). Shapley values, which originate from game theory, have several desirable properties which make them optimal for evaluating features contribution in machine-learning. Formally, a Shapley value is the average marginal contribution of a feature value across all possible combinations of features. The SHAP contribution of a given feature was computed by first summing the absolute Shapley values on all epochs and then averaging across all sleep stages, thus leading to a single importance score for each feature.”

Reviewer #2 (Recommendations for the authors):1. As mentioned in the previous section, I am very surprised that the work of Stephansen et al., (Stephansen et al., Nature Communications 2018, doi: 10.1038/s41467-018-07229-3) is not cited. It is extremely relevant as Stephansen et al., also used a large amount of public data to train and test an automated sleep scoring algorithm (based on neural networks). They are very similar ideas developed in this article (for example they also introduced the use of probabilistic scoring with hypnodensities). Some ideas could also be worth implementing with the proposed algorithm such as testing the impact of common sleep disturbances or shortening the epoch window. Finally, most if not all the data used in Stephansen et al., are available on the SleepData.org platform (used by the authors) so the authors could directly benchmark the performance of the two approaches. From my personal experience, Stephansen algorithm has three disadvantages compared to the present algorithm: (i) it is slower and heavier, (ii) it requires python machine-learning packages such as TensorFlow, which are frequently updated, (iii) it is very difficult to modify and understand how it works (black box). I think it would be interesting for the authors to show how their approach could mitigate these issues.

This was a great idea. Towards that end, we have now done so, and describe the details of this extensive revision in point 1 of the Essential Revisions.

2. The tool would greatly benefit from a graphical interface to install and use the software. Another interesting function could be to process a list of EDF files (for example, all the EDF files in a given folder). From personal experience again, an automated sleep scoring algorithm will not be used in clinical settings at a large scale without a GUI.

We indeed agree that the absence of graphical user interface (GUI) may prevent wide adoption of our software, especially in clinical settings. We note, however, that designing and maintaining a cross-platform GUI is virtually impossible without dedicated funding and/or a full-time software engineer. While it is therefore unlikely that YASA will have a GUI in the near future, we have made efforts to integrate the outputs of the algorithm with existing sleep GUIs. For instance, the sleep scores can be loaded and edited in several free GUIs, such as EDFBrowser, Visbrain (Python) or SleepTrip (Matlab). We have now added this point in the “Limitations and future directions” subsection of the Discussion:

“Second, an obstacle to the wide adoption of the current algorithm is the absence of a graphical user interface (GUI). Designing and maintaining an open-source cross-platform GUI is in itself a herculean task that requires dedicated funding and software engineers. Rather, significant efforts have been made to facilitate the integration of YASA’s outputs to existing sleep GUIs. Specifically, the online documentation of the algorithm includes examples on how to load and edit the sleep scores in several free GUIs, such as EDFBrowser, Visbrain and SleepTrip.”

Please see Author response image 2 for a screenshot of the online documentation.

Furthermore, we have now added a code snippet illustrating how to run the sleep scoring on all the EDF files present in a folder at once. This is now mentioned in the “Software implementation” subsection of the Results:“The general workflow to perform the automatic sleep staging is described below. In addition, we provide code snippets showing the simplest usage of the algorithm on a single EDF file (Figure S8) or on a folder with multiple EDF files (Figure S9).”

3. I think the algorithm will be closer to the current consensus if the smoothing was restricted to preceding and not following epochs. Also, could the authors examine the impact of the duration of the smoothing period on performance? A too large smoothing might decrease the performance for transitory stages such as N1.

This has been addressed in point 6 of the Essential Revisions (“Temporal Smoothing” subsection).

4. Since the proposed algorithms can provide a probabilistic score, it would be interesting to explore further the differences between the human gold-standard scores and the algorithm. For example, for misclassified epochs, is the confidence lower in all stages? In the case of an error, is the second-highest probability typically the correct one?

Great questions. We refer the reviewer to point 2 of the Essential Revisions where we address both.

Related to the same point, when writing "for epochs ﬂagged by the algorithm as high-conﬁdence ({greater than or equal to}80% conﬁdence) than in epochs with a conﬁdence below 80%", can the authors indicate the % of epochs {greater than or equal to} and < to 80% confidence? Could the authors show the distribution of these confidences per sleep stage?

We have now added a supplementary figure showing the distribution of confidence levels for each stage:

“Similarly, and as expected, accuracy was significantly greater for epochs flagged by the algorithm as high-confidence (≥80% confidence) than in epochs with a confidence below 80% (95.90 ± 3.70% and 62.91 ± 6.61% respectively, p<0.001). The distribution of confidence levels across sleep stages is shown in Figure 1—figure supplement 2. The algorithm was most confident in epochs scored as wakefulness by the human scorer (mean confidence across all epochs = 92.7%) and least confident in epochs scored as N1 sleep (mean = 63.2%).”

5. In the "Descriptive Statistics" section, could the authors add a table assessing the difference between the training and test sets for the variables mentioned in the text?

In the revision, we now provide a new table (Table 1) that includes demographics/health data of the training and testing sets as well as statistical comparison between the two.

Furthermore, we have now added the following paragraph in the Results section:

“Demographics and health data of the testing set 1 are shown in Table 1. Importantly, although participants were randomly assigned in the training/testing sets, there were some significant differences between these two sets. Specifically, age and BMI were lower in the testing set compared to the original training set (p’s < 0.05), although the effect sizes of these differences were all below 0.17 (considered small or negligible). Importantly, there was no significant difference in the sex ratio, race distribution, or in the proportion of individuals diagnosed with insomnia, depression or diabetes. Furthermore, there was no difference in AHI and the proportion of individuals with minimal, mild, moderate or severe sleep apnea.”

6. How was the AHI obtained? Was it provided for each dataset and individual or computed by the authors?

The AHI was pre-calculated and provided with each dataset on the NSRR website. Specifically, we used the “ahi_a0h3” variable, which is the Apnea-Hypopnea Index (AHI) >= 3% – number of [all apneas] and [hypopneas with >= 3% oxygen desaturation] per hour of sleep. This has now been clarified in the Methods section of the manuscript:

“Demographics and health data such as the age, sex, race/ethnicity, body mass index (BMI), apnea-hypopnea index (AHI, 3% desaturation), as well as medical diagnosis of insomnia, depression, diabete, hypertension were also provided for each dataset.”

7. "These results match recently developed deep-learning-based algorithms for automatic sleep staging (Guillot et al., 2020; Perslev et al., 2021)." Citing the Stephansen paper is important here, especially since the authors could apply their algorithms to the same data. Also, the Stephansen studies should be cited when discussing the advantage of a probabilistic scoring (they coined the term hypnodensity).

We have now added the Stephansen 2018 citation throughout the manuscript (see also point 1 of the Essential Revisions).

8. Out of curiosity, could the list of features used by the algorithms be changed by future users? This could be an interesting feature for future developments.

We refer the reviewer to point 8 of the Essential Revisions.

Reviewer #3 (Recommendations for the authors):In this study, Vallat and Walker describe a new sleep scoring tool that is based on a classification algorithm using machine-learning approaches in which a set of features is extracted from the EEG signal. The algorithm was trained and validated on a very large number of nocturnal sleep datasets including participants with various ethnicities, age and health status. Results show that the algorithm offers a high level of sensitivity, specificity and accuracy matching or sometimes even exceeding that of typical interscorer agreement. Importantly, a measure of the algorithm's confidence is provided for each scored epoch in order to guide users during their review of the output. The software is described as easy to use, computationally low-demanding, open source and free.This paper addresses an important need in the field of sleep research. There is indeed a lack of accurate, flexible and open source sleep scoring tools. I would like to commend the authors for their efforts in providing such a tool for the community and for their adherence to the open science framework as the data and codes related to the current manuscript are made available. I predict that this automated tool will be of use for a large number of researchers in the field. I also enjoyed reading the paper that is nicely written. However, I have some concerns listed below that need to be addressed and comments that might help improving the overall quality of the paper.

– There are some overstatements that need to be toned down.

– In the title, the word "universal" should be removed. The tool was trained and validated on nocturnal sleep data. Sleep characteristics (eg duration and distribution of sleep stages etc.) are different, for example, during diurnal sleep (nap) and the algorithm might not perform as well on nap data. Note that there is a large number of scientific studies in which diurnal, instead of nocturnal, sleep paradigms are used as these paradigms are easier to implement in lab settings. The algorithm being optimized for nocturnal sleep (with eg the definition of specific sleep stage weights that are specific to nocturnal recordings), it is unknown how it would perform on nap data for example.

The word “universal” has now been removed from the title. Thanks for that point. Furthermore, we have added a “Limitations and future directions” subsection in the Discussion, which includes among others the fact that the algorithm was trained and validated only on nocturnal data, and as such may not be as accurate for daytime naps data:

“Third, the algorithm was exclusively tested and evaluated on full nights PSG recordings, and as such its performance on shorter recordings such as daytime naps is unknown. Further testing of the algorithm is therefore required to validate the algorithm on naps. While YASA can process input data of arbitrary length, performance may be reduced on data that are shorter than the longest smoothing windows used by the algorithm (i.e. 7.5 minutes).”

– In the abstract, the following sentence needs to be altered: "This tool offers high sleep-staging accuracy matching or exceeding human accuracy". Exceeding human accuracy might be misleading as human scores are used as the ground-truth in the validation process. The algorithm exceeds the accuracy of some human scorers and matches the scores of the best scorer.

The abstract has been edited as follow:

“This tool offers high sleep-staging accuracy matching human accuracy and interscorer agreement no matter the population kind.”

– Acknowledgement of – and comparisons to – already available tools in the fieldThere are plenty of automated sleep scoring tools available in the field (most of them are not open source and rather expensive though – as noted by the authors). A short review of the existing tools, their performance and accessibility would be a welcomed addition to the introduction. In line with this comment, other published algorithms were tested on the same set of data used to train and validate the present algorithm. The authors refer to their work for comparison between algorithms but it would be a very nice addition to the paper to provide (and test for) such comparisons. It is currently unclear whether the present algorithm performs any better than algorithms already available in the field.

This has been addressed in point 1 of the Essential Revisions.

– Further improvementsThe algorithm performs worse on N1 stage, older individuals and patients presenting sleep disorders (sleep fragmentation). It would be helpful to add a section to the manuscript with some considerations on how accuracy could be improved for these different issues. Eg, should one consider training the algorithm on older datasets in order to improve accuracy of scoring when studying aging? Same applies to sleep disorders. It is currently unclear whether the variety of datasets used to train the algorithm is beneficial in these cases.

As discussed in point 3 of the Essential Revisions, we have now added a “Limitations and future directions” subsection in the Discussion to present several ideas for improving the algorithm, especially in sleep-disordered patients.

In the same vein, as algorithms usually perform poorly on fragmented sleep data (as acknowledged by the authors), it would be interesting to present the results of the validation test separately for the healthy volunteers (N-25) and the patients (N=55).

The performance of the algorithm on the DOD testing set is now reported separately for healthy adults and patients with sleep disorders (point 4 of the Essential Revisions).

– Justification for some methodological choices– Nights were cropped to 15 minutes before and after sleep to remove irrelevant extra periods of wakefulness or artefacts on both ends of the recording. This represents an issue for the computation of important sleep measures such as sleep efficiency and latency as the onset/offset of sleep might be missed.

This truncation has now been removed from the pipeline and all the results have been updated accordingly. Thanks for that.

– How were features selected? A description of the features needs to be provided.

We have edited the “Features extraction” subsection of the Methods as follow:

“These features were selected based on prior work in features-based classification algorithms for automatic sleep staging (Krakovská and Mezeiová 2011; Lajnef et al., 2015; Sun et al., 2017). For example, it was previously reported that the permutation entropy of the EOG/EMG and the EEG spectral powers in the traditional frequency bands are the most important features for accurate sleep staging (Lajnef et al., 2015), thus warranting their inclusion in the current algorithm. Several other features are derived from the authors’ own works with entropy/fractal dimension metrics^1^. Importantly, the features included in the current algorithm were chosen to be robust to different recording montages. As such, we did not include features that are dependent on the phase of the signal, and/or that require specific events detection (e.g. slow-waves, rapid eye movements). However, the time-domain features are dependent upon the amplitude of the signal, and sleep staging may fail if the input data is not expressed in standard unit (uV) or has been z-scored prior to applying the algorithm.”

^1^ https://github.com/raphaelvallat/antropy

Which features were Z scored exactly and why not all?

The full model includes both the original features in raw units (no smoothing or normalization), as well as the normalized-smoothed version of these features. This has now been clarified in the manuscript:

“To take this into account, all the smoothed features were then z-scored across each night, i.e. expressed as a deviation from the night’s average. The inclusion of such normalized features aids in accommodating the error-potential impact of inter-individual variability upon the algorithm, and thus improves final accuracy. The final model includes the 30-sec based features in original units (no smoothing or scaling), as well as the smoothed and normalized version of these raw features. The raw features were included to increase the temporal specificity and keep absolute values that can be compared across individuals regardless of inter-individual variability.”

– The custom sleep stage weights procedure is unclear. Why was only a sub-sample used to determine best weights?

Subsampling was initially performed to improve speed of the parameter search, but to address this concern, we now use the full training set to find the best class weights. In addition, we now provide more details on the procedure in the Methods section:

“The best weights were found by running a cross-validated parameter search on the full training set, with the average of the accuracy and F1-scores as the optimization metric. A total of 324 possible combinations of class weights were tested. The parameter space was defined as the cartesian product of N1: [1.6, 1.8, 2, 2.2], N2: [0.8, 0.9, 1], N3 / REM / Wake: [1, 1.2, 1.4]. The best weights were: 2.2 for N1, 1 for N2 and Wake, 1.2 for N3 and and 1.4 for REM. Python code for the grid search of the best class weights can be found here”.

– Was the smoothing done with 5min or 15min rolling average? While it is critical to include this temporal window, it looks rather wide. Human scorers usually don't go that far back when scoring but usually just a few epochs (ie 2-3 which is 1min30). What was the rational for choosing such a wide temporal window?

We apologize for the confusion caused by a typographical error in the previous version of our manuscript. We did not use a 15 min rolling average. Instead, temporal smoothing was previously performed using two distinct rolling windows, i.e. 1) a 5.5-minutes centered (symmetric, 11 epochs), triangular-weighted rolling average, and 2) a rolling average of the last 5 minutes (10 epochs) prior to the current epoch. As requested by the reviewer, we have now added an extensive comparison of the impact of the length of the rolling window on the overall performance of the algorithm, which led us to update the length of the rolling windows. A full summary can be found in point 6 of the Essential Revisions (“Temporal smoothing”).

– It is currently unclear when / how the EEG and EMG data were analyzed.

As indicated in earlier comments, the “Features extraction” subsection of the Methods has been extensively reworked in order to clarify some details on PSG signal preprocessing and features computation.